# PKCβII activation requires nuclear trafficking for phosphorylation and Mdm2-mediated ubiquitination

Xiao Min, Shujie Wang, Xiaohan Zhang, Ningning Sun, Kyeong-Man Kim

**PKCβII, a conventional PKC family member, plays critical roles in the regulation of a variety of cellular functions. Here, we employed loss-of-function approaches and mutants of PKCβII with altered phosphorylation and protein interaction behaviors to identify the cellular mechanisms underlying the activation of PKCβII. Our results show that 3-phosphoinositide–dependent protein kinase-1 (PDK1)–mediated constitutive phosphorylation of PKCβII at the activation loop (T500) is required for phorbol ester–induced nuclear entry and subsequent Mdm2-mediated ubiquitination of PKCβII, whereas ubiquitination of PKCβII is required for the PDK1-mediated inducible phosphorylation of PKCβII at T500 in the nucleus. After moving out of the nucleus, PKCβII interacts with actin, undergoes inducible mTORC2-mediated phosphorylation at the turn motif (T641), interacts with clathrin, and then translocates to the plasma membrane. This overall cascade of cellular events intertwined with the phosphorylation at critical residues and Mdm2-mediated ubiquitination in the nucleus and along with interactions with actin and clathrin plays roles that encompass the core processes of PKC activation.**

## Introduction

PKC is a family of protein kinases that exert functional control over other proteins through the phosphorylation of serine and threonine residues (Inoue et al, 1977; Takai et al, 1977). Based on their second messenger requirements, PKCs are divided into three subfamilies, conventional, novel, and atypical (Kikkawa et al, 1989; Newton, 1995).

For the activation of conventional PKCs that include PKCβII, increases in the concentration of DAG and $Ca^{2+}$, which bind to the C1 and C2 region (Takai et al, 1979a, 1979b; Giorgione et al, 2003), respectively, play critical roles. In addition, the involvement of phosphorylation occurring at three critical residues while PKC is shuttling between the plasma membrane and the cytosol is suggested to play important roles in the activation (Newton, 2001). PKCβII is phosphorylated in the cytosol at three phosphorylation sites shortly after being synthesized: at an activation loop (A-loop) by the phosphoinositide-dependent kinase (PDK-1), and then at two positions in the COOH terminus—the turn motif (TM) and the hydrophobic motif (HM) (Newton, 2001; Freeley et al, 2011). The phosphorylation of PKC at three conserved sites can occur constitutively or inducibly.

Constitutive phosphorylation has been described as a priming event that places PKC in the cytosol in an inactive but signaling-competent state so that it is ready to be activated by lipid second messengers such as DAG (Flint et al, 1990; Newton, 2018). For the primed or mature PKC in a non/hypo-phosphorylation state, cellular stimulation results in the recruitment of PKC to the plasma membrane and the additional (inducible) phosphorylation at one or more sites such as the A-loop and TM. Inducible phosphorylation is generally believed to be an activation process of PKC (Freeley et al, 2011). Here, we set out to investigate the roles of constitutive and inducible phosphorylation in the activation of PKCβII and whether and how they are functionally or mechanistically interconnected.

Along with phosphorylation, a recent study has shown that PKCβII is also regulated through Mdm2-mediated ubiquitination (Min et al, 2019). In addition to proteasomal degradation of the target protein, ubiquitination regulates various cellular functions, such as gene expression, cell signaling, and intracellular trafficking (Miranda & Sorkin, 2007; Komander & Rape, 2012). Similarly, Mdm2-mediated ubiquitination of PKCβII is known to be responsible for the down-regulation and activation of PKCβII, as determined by its translocation to the plasma membrane (Min et al, 2019).

Actin, the most abundant and highly conserved protein that engages in multiple protein–protein interactions, is known for its variety of functional interactions with PKC. For example, filamentous actin is a principal anchoring protein for PKCε within intact nerve endings (Prekeris et al, 1996). PKCβII also interacts with the cytoskeleton during PMA-induced translocation of PKCβII from the cytosol to the plasma membrane

Department of Pharmacology, College of Pharmacy, Chonnam National University, Gwang-Ju, Republic of Korea

Correspondence: kmkim@jnu.ac.kr
Xiaohan Zhang's present address is School of Pharmaceutical Sciences, Guizhou University, Guiyang, China

(Pascale et al, 2004). In contrast, PKCβI, which exhibits a high sequence homology with PKCβII, does not bind to actin. The isozyme specificity of PKC in the interaction with filamentous actin is known to result from divergent phorbol ester and calcium dependencies (Slater et al, 2000). Interestingly, the interaction of PKCβII with actin results in marked enhancement of autophosphorylation of PKCβII (Blobe et al, 1996), suggesting that actin could be involved in the regulation of PKC phosphorylation.

The association between actin filament and clathrin-mediated endocytosis (CME) has long been proposed (Salisbury et al, 1980): F-actin dynamics are required for multiple stages of clathrin-coated vesicle formation (Yarar et al, 2005), and actin assembly generally precedes dynamin 2 recruitment during the late phases of CME (Grassart et al, 2014). In addition, it has been shown that both actin and clathrin are involved in the recruitment of PKCβII to the plasma membrane (Blobe et al, 1996; Min et al, 2019).

Based on this experimental evidence, in the current study, we set out to clarify the molecular mechanisms involved in the activation of PKCβII. To that end, we determined the mechanistic and functional relationships between ubiquitination and constitutive or inducible phosphorylation at the three major motifs, focusing on interactions with actin and clathrin.

# Results

## Actin is involved in the PMA-induced translocation of PKCβII to the plasma membrane

PKCβI and PKCβII have identical amino acids, up to 621, but about 50 amino acids in the C-terminus are disparate. However, they possess distinct functional and regulatory properties; for example, PKCβII but not PKCβI undergoes Mdm2-mediated ubiquitination (Min et al, 2019), and PKCβII but not PKCβI potentiates the PMA-induced endocytosis of G protein-coupled receptors (GPCRs) such as dopamine $D_3$ receptor ($D_3$R) (see Fig S1A), which is known to undergo PKC-mediated endocytosis (Cho et al, 2007). These findings are in agreements with a previous report that showed that PKCβII but not PKCβI translocates from the cytosol to the plasma membrane in response to PMA treatment (Pascale et al, 2004). In particular, the translocation of PKCβII has been found to not be associated with synchronous RACK1 relocation but to require filamentous actin (Blobe et al, 1996; Pascale et al, 2004). In agreement with this report, we also found that the interaction of PKCβII, not PKCβI, with actin increased after PMA treatment (Fig S1B).

To further characterize the roles of actin in PKCβII-mediated endocytosis of $D_3$R, cells were pretreated with latrunculin A (LatA), which sequesters G-actin to suppress actin polymerization (Braet et al, 1996). As shown in Fig S1C, PMA-induced translocation of PKCβII was blocked by LatA pretreatment.

CME and caveolar endocytosis are two representative endocytic routes of GPCRs, which display distinct endocytic properties

and sensitivities to endocytic inhibitors (Ivanov, 2008; Guo et al, 2015). Here, PMA-induced endocytosis of $D_3$R was significantly inhibited when the endogenous clathrin heavy chain (CHC) was knocked down; in contrast, knockdown of cellular caveolin1 (Cav1) did not affect it, suggesting that $D_3$R endocytosis occurred via CME. The inhibitory activities of LatA on the $D_3$R endocytosis were abolished in CHC-KD cells but remained intact in control knockdown (Con-KD) cells and Cav1-KD cells (Fig S1D), suggesting that actin mediates the CME of $D_3$R.

To confirm the functional roles of the interactions between PKCβII and actin in the translocation of PKCβII, we created a PKCβII mutant that lacks an actin binding site. A previous study had shown that the ActX1 and ActX2 regions (shown in Fig 1A), which are homologous to the acting binding sites of troponin I, could be responsible for binding with actin (Blobe et al, 1996). Based on these findings, we prepared ActX1-PKCβII and ActX2-PKCβII constructs separately and then combined them to create ActX-PKCβII. As shown in Fig 1B, ActX-PKCβII did not engage in interactions with actin. In addition, ActX-PKCβII failed to translocate toward the plasma membrane (Fig 1C), and the endocytic activities of PKCβII were increasingly diminished with each mutation (Fig 1D).

## Clathrin interacts with PKCβII to mediate translocation to plasma membrane

Because actin is involved in the CME of $D_3$R, we further characterized the functional relationship between PKCβII and clathrin. In response to PMA treatment, PKCβII translocated toward the plasma membrane in Con-KD cells and Cav1-KD cells but remained in the cytosolic region of CHC-KD cells (Fig 2A, middle panel). The roles of clathrin in the regulation of PKCβII functions were further confirmed with a mutant of PKCβII in the clathrin-binding site ($^{277}$LLSQE$^{281}$) (Lafer, 2002) CHCX-PKCβII. This mutant neither interacted with clathrin after PMA treatment (Fig 2B) nor did it translocate to the plasma membrane (Fig 2C) and failed to mediate $D_3$R endocytosis (Fig 2D). These results suggest that interaction with clathrin is required for the proper functioning of PKCβII, including translocation to the plasma membrane and GPCR endocytosis.

## Neither clathrin nor actin is needed for PDK1-dependent Mdm2-mediated ubiquitination of PKCβII

A recent study showed that Mdm2-mediated ubiquitination of PKCβII, whose original role is the degradation of target protein, is required for the PMA-induced translocation to the plasma membrane (Min et al, 2019). Because clathrin and actin are needed for the PKCβII translocation, we assessed their roles in the ubiquitination of PKCβII. As shown in Fig 3A and B, knockdown of CHC or a mutation in the clathrin binding site on the PKCβII did not affect PKCβII ubiquitination (Fig 3A and B) and neither did a mutation in the actin binding site (Fig 3C).

Like other isotypes of PKCs that belong to conventional and novel families, PKCβII is functionally regulated by specific phosphorylations, including those at three critical sites located in

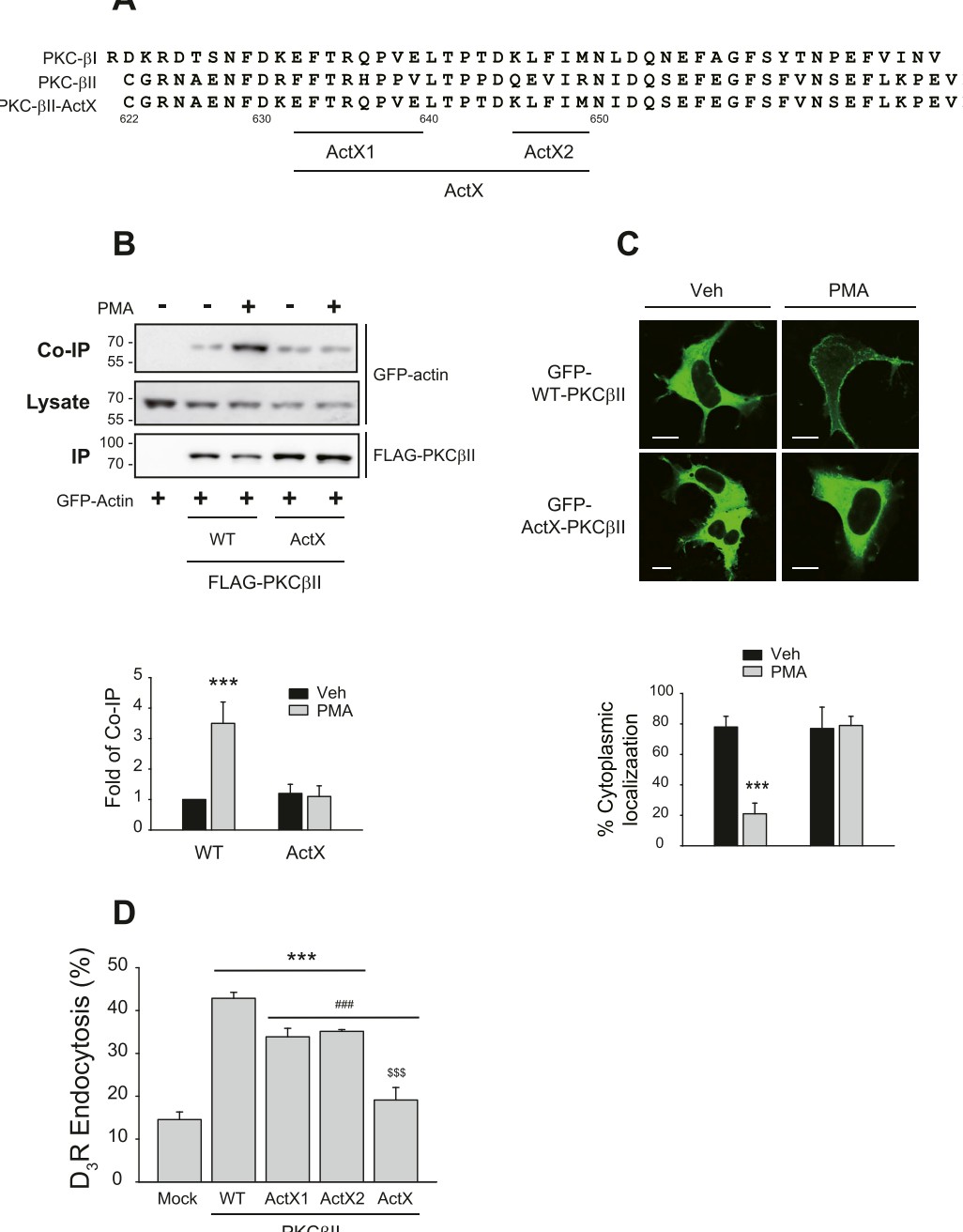

**Figure 1. Interaction with actin is needed for the PMA-induced translocation of PKCβII to the plasma membrane.**
**(A)** Comparison of amino acid sequences in the carboxy terminal tail of PKCβI and PKCβII. ActX1 and ActX2 represent putative actin binding domains, which are exclusively found on PKCβII. **(B)** HEK-293 cells were transfected with GFP-tagged actin along with FLAG-tagged WT-PKCβII or ActX-PKCβII, in which some of the amino acids located on both ActX1 and ActX2 domains are mutated. The cells were treated with 100 nM PMA for 5 min. Cell lysates were immunoprecipitated with anti-FLAG-agarose beads. Co-IP/lysates and IPs were immunoblotted with antibodies against GFP and GFP, respectively. **(C)** HEK2-93 cells were transfected with GFP-tagged WT- or ActX-PKCβII. Cells were treated with vehicle or 100 nM PMA for 15 min. ***P < 0.001 compared with other groups (n = 5). Horizontal bars represent 10 μm. **(D)** HEK-293 cells expressing $D_3R$ (2.4 pmol/mg protein) were transfected with mock, WT-, ActX1-, ActX2-, or ActX-PKCβII, followed by treatment with 100 nM PMA for 30 min. ***P < 0.001 compared with the mock group; ###P < 0.001 compared with the ActX group; $$$P < 0.001 compared with other PKCβII groups (n = 3).

the activation loop (T500), TM (T641), and HM (S660) (Freeley et al, 2011; Newton, 2018). Because PDK1 mediates the constitutive phosphorylation of T500 of PKCβII in the basal state (Dutil et al, 1998; Le Good et al, 1998), which is needed for the subsequent phosphorylation at T641 (Balendran et al, 2000), we also assessed the roles of PDK1 in the PMA-induced ubiquitination of PKCβII. As shown in Fig 3D, the ubiquitination of PKCβII was abolished in PDK1-KD cells.

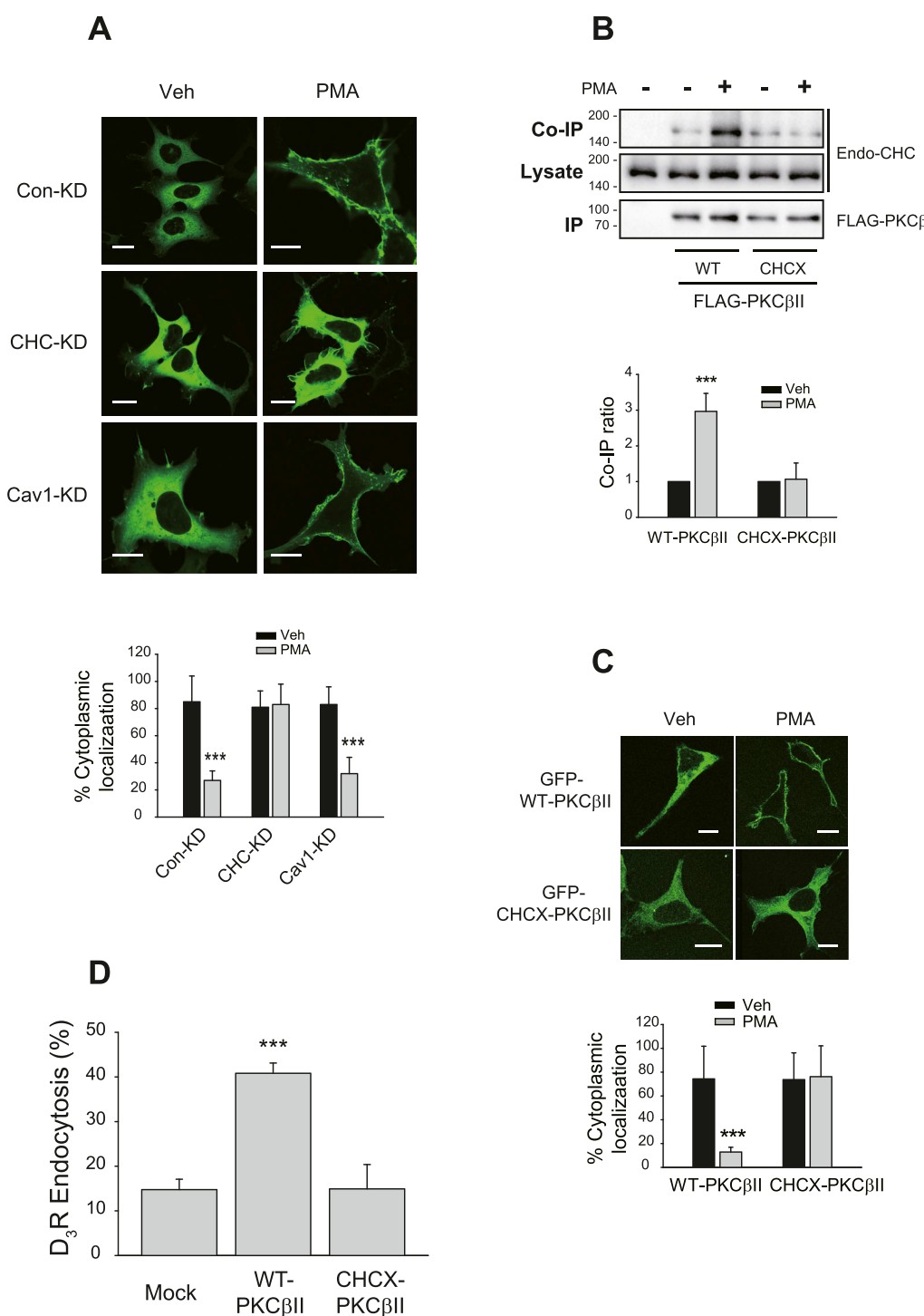

**Figure 2. Interaction with clathrin is required for the PMA-induced translocation of PKCβII to the plasma membrane.**
**(A)** Con-KD, CHC-KD, and Cav1-KD HEK-293 cells were transfected with GFP-PKCβII. Cells were treated with vehicle or 100 nM PMA for 15 min. Knockdown efficiencies for CHC and Cav1were 90–95%. ***$P$ < 0.001 compared with corresponding vehicle-treated groups (n = 5). Horizontal bars represent 10 $\mu m$. **(B)** HEK-293 cells were transfected with FLAG-tagged WT or CHCX-PKCβII, in which clathrin binding sites are mutated, followed by treatment with 100 nM PMA for 5 min. Cell lysates were immunoprecipitated with anti-FLAG agarose beads. Co-IP/lysates and IPs were immunoblotted with antibodies against clathrin heavy chain and FLAG, respectively. ***$P$ < 0.001 compared with other groups (n = 3). **(C)** HEK-293 cells were transfected with GFP-tagged WT or CHCX-PKCβII, followed by treatment with 100 nM PMA for 15 min. ***$P$ < 0.001 compared with other groups (n = 5). Horizontal bars represent 10 $\mu m$. **(D)** HEK-293 cells were transfected with D$_3$R along with mock vector, WT, or CHCX-PKCβII. The cells were treated with 100 nM PMA for 30 min. ***$P$ < 0.001 compared with other groups (n = 3).

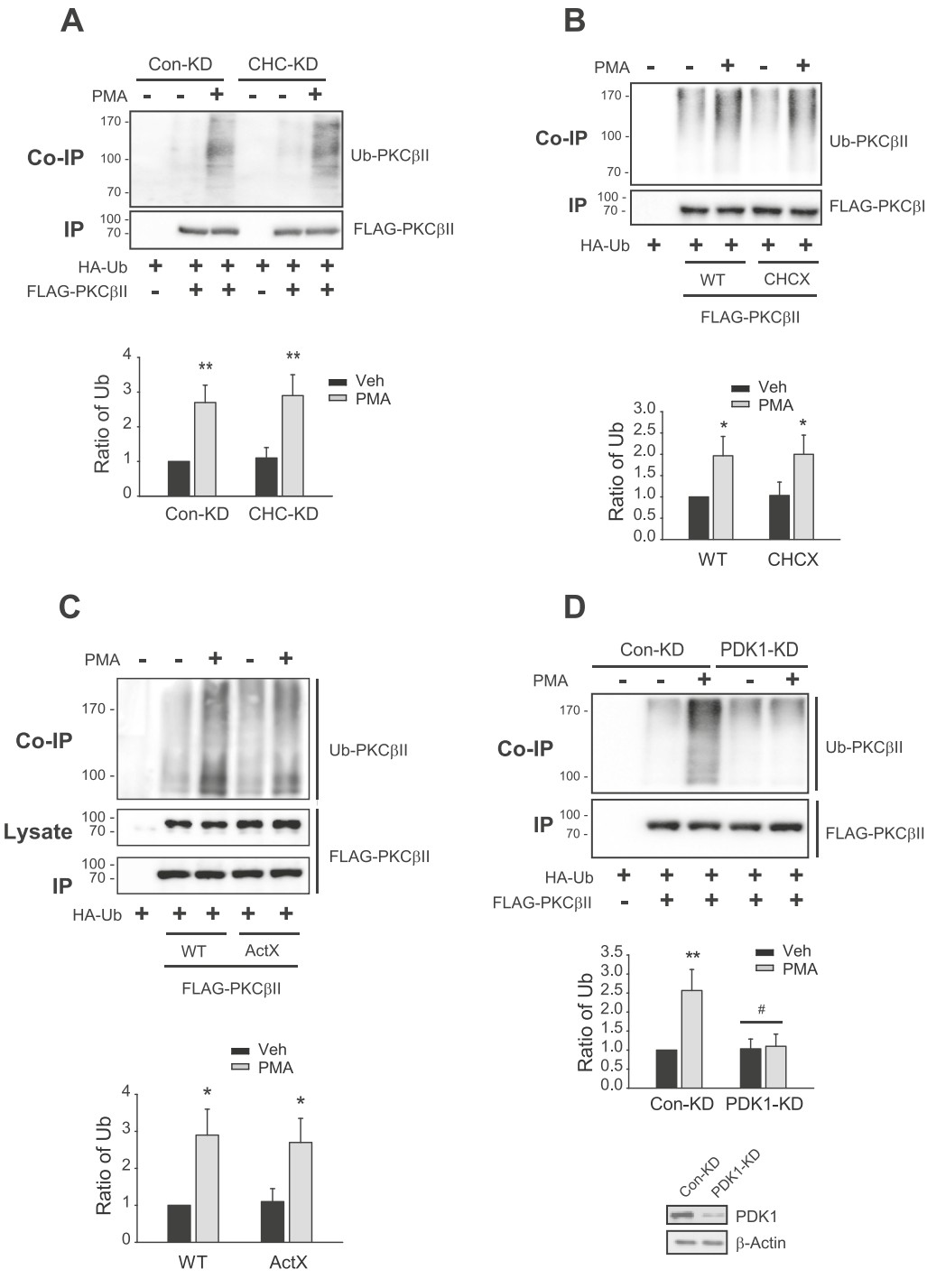

**Figure 3. Clathrin and actin are not needed for Mdm2-mediated ubiquitination of PKCβII.**
Cells were transfected with HA-Ub and FLAG-tagged PKCβII constructs. The cells were treated with 100 nM PMA for 15 min. Cell lysates were immunoprecipitated with anti-FLAG agarose beads. Co-IP/lysates and IPs were immunoblotted with antibodies against HA and FLAG, respectively, to detect ubiquitinated and unmodified PKCβII. **(A)** Con-KD and CHC-KD HEK-293 cells were transfected with HA-Ub and FLAG-PKCβII. **P < 0.01 compared with each vehicle groups (n = 3). **(B)** HEK-293 cells were transfected with HA-Ub and FLAG-tagged WT- or CHCX-PKCβII. *P < 0.05 compared with each vehicle group (n = 3). **(C)** HEK-293 cells were transfected with HA-Ub and FLAG-tagged WT- or ActX-PKCβII. *P < 0.05 compared with corresponding vehicle-treated groups (n = 3). **(D)** Con-KD and PDK1-KD HEK-293 cells were transfected with HA-Ub and FLAG-PKCβII. **P < 0.01 compared with other groups; #P < 0.05 compared with the PMA/Con-KD group (n = 3). Knockdown efficiency for PDK1 was about 90%.

These results suggest that PDK1-mediated constitutive phosphorylation of PKCβII at T500 is likely to be involved in the ubiquitination of PKCβII; in turn, ubiquitination occurs at upstream levels of actin or clathrin, allowing PKCβII to translocate to the plasma membrane.

## PDK1-mediated phosphorylation of PKCβII at T500 initiates the activation processes of PKCβII that include ubiquitination and translocation to the plasma membrane

Along with ubiquitination (Fig 3D), PDK1 was also required for the constitutive phosphorylation of PKCβII at T641 (Fig 4A) and the PMA-induced translocation to the plasma membrane (Fig 4B). Among the three critical phosphorylation sites (T500, T641, S660), a mutation in T500, which is phosphorylated by PDK1, inhibited the translocation of PKCβII to the plasma membrane (Fig 4C). In addition, the endocytosis of D₃R was enhanced by co-expression of WT- or T641A-PKCβII but not T500A-PKCβII (Fig 4D). These results overall suggest that the constitutive phosphorylation of PKCβII at T500 controls the constitutive phosphorylation of PKCβII at T641 and other PMA-induced cellular events that include ubiquitination and translocation to the plasma membrane.

## Constitutive phosphorylation of PDK1 at T500 is required for the PMA-induced nuclear entry of PKCβII

As expected from the roles of PKCβII ubiquitination in the translocation to the plasma membrane (Min et al, 2019), PMA-induced ubiquitination was observed in the cells expressing WT- or T641A/S660A-PKCβII but not in the cells expressing T500A-PKCβII (Fig 5A and B).

Mdm2-mediated ubiquitination of PKCβII at position 63 (K63, Fig S2) occurs in the nucleus (Min et al, 2019), and PKCβII needs to enter the nucleus to be ubiquitinated. As predicted from the behaviors of PKCβII mutants during translocation and ubiquitination, all PKCβII mutants except T500A-PKCβII were able to enter the nucleus in response to PMA treatment when it was determined by immunocytochemistry (Fig 5C) or by subcellular fractionation (Fig S3).

These results suggest that PDK1-mediated constitutive phosphorylation of PKCβII at T500 is a prerequisite for the PMA-induced nuclear entry that determines the Mdm2-mediated ubiquitination in the nucleus and the translocation of ubiquitinated PKCβII to the plasma membrane.

## Interaction of PKCβII with clathrin or actin, required for the translocation of PKCβII, occurs in ubiquitination-dependent manner

Because interactions with clathrin and actin are needed for the translocation of PKCβII to the plasma membrane (Figs 1C and 2A) but not for its ubiquitination (Fig 3A–C), it is likely that the ubiquitination of PKCβII is needed for interactions with clathrin or actin. Indeed, 2KR-PKCβII, a ubiquitination-deficient mutant of PKCβII in which K668 and K672 are mutated and cannot be ubiquitinated (Min et al, 2019), significantly weakly interacted with clathrin than the wild-type PKCβII did (Fig 6A). In addition, the interaction between PKCβII and clathrin was significantly reduced in Mdm2-KD cells compared with Con-KD cells (Fig 6B). Interestingly, clathrin translocated to the plasma membrane in response to PMA treatment (Fig 6C), suggesting that PKCβII and clathrin might act as vehicles and provide a driving force to recruit each other to the plasma membrane after PMA treatment.

Treatment with the Mdm2 inhibitor (Fig S4A) or mutations in the ubiquitination sites on PKCβII (Fig S4B) also inhibited the interaction with actin. These results overall suggest that Mdm2-mediated ubiquitination of PKCβII occurs before PKCβII interacts with clathrin or actin.

## Inducible phosphorylation of PKCβII at T500 and T641 occurs in the nucleus and cytosol, respectively

Phosphorylation of PKC at three conserved phosphorylation sites can occur constitutively and/or in an inducible manner: the former supposedly is a pre-required step for the catalytic activation of PKC, and the latter is believed to represent an activation process (Freeley et al, 2011). Because the ubiquitination and the interactions with actin and clathrin were required for the PMA-induced translocation to the plasma membrane, we wanted to determine whether these are related to the phosphorylation during the activation process.

It is known that PDK1-mediated phosphorylation of PKCβII at T500, followed by the dissociation of PDK1 from PKCβII, is needed for the phosphorylation of PKCβII at T641 (Freeley et al, 2011). We assessed the time course of the interaction between PKCβII and PDK1 and found that their interactions increased shortly (1 min) after PMA treatment and that the dissociation started at around 5 min (Fig 7A).

We then assessed where in the subcellular domain the PMA-induced interaction and dissociation occurs. To that end, cells were treated with leptomycin B (LMB), which inhibits exportin1, the major karyopherin receptor that mediates the nuclear export of a wide range of proteins. PMA-induced interactions between PDK1 and PKCβII were enhanced at 1 min (Fig 7B), but the dissociation at 5 min was abolished (Fig 7C) when the cells were pretreated with LMB. These results overall suggest that PMA-induced interactions between PDK1 and PKCβII at 1 min, which determine the inducible phosphorylation of PKCβII at T500, occur in the nucleus. In contrast, the PMA-induced dissociation between the two proteins at 5 min that determines the phosphorylation of PKCβII at T641 occurs after PKCβII translocates to the cytoplasm.

## Mdm2-mediated ubiquitination is needed for the inducible phosphorylation of PKCβII at T500

Because Mdm2-mediated ubiquitination and interaction with actin and clathrin are needed for the translocation of PKCβII to the plasma membrane, we assessed the roles of ubiquitination in the inducible phosphorylation of PKCβII. As shown in Fig 8A, the PMA-induced increase in the phosphorylation of PKCβII at T500 and T641 at 1 min was significantly inhibited by knockdown of Mdm2. In addition, the PMA-induced dissociation between PKCβII and PDK1 at 5 min was inhibited in Mdm2-KD cells (Fig 8B). In agreement with these results, 2KR-PKCβII, which cannot be ubiquitinated, showed the same pattern that it exhibited in Mdm2-KD cells (Fig 8C and D). These results suggest that Mdm2-mediated ubiquitination of PKCβII is a prerequisite for the stimuli-induced interaction between PDK1 and PKCβII in the nucleus.

## mTORC2-mediated inducible phosphorylation of PKCβII at T641 requires interaction with actin

According to our results, the constitutive phosphorylation of PKCβII at T500 that occurs before PKCβII enters the nucleus is a

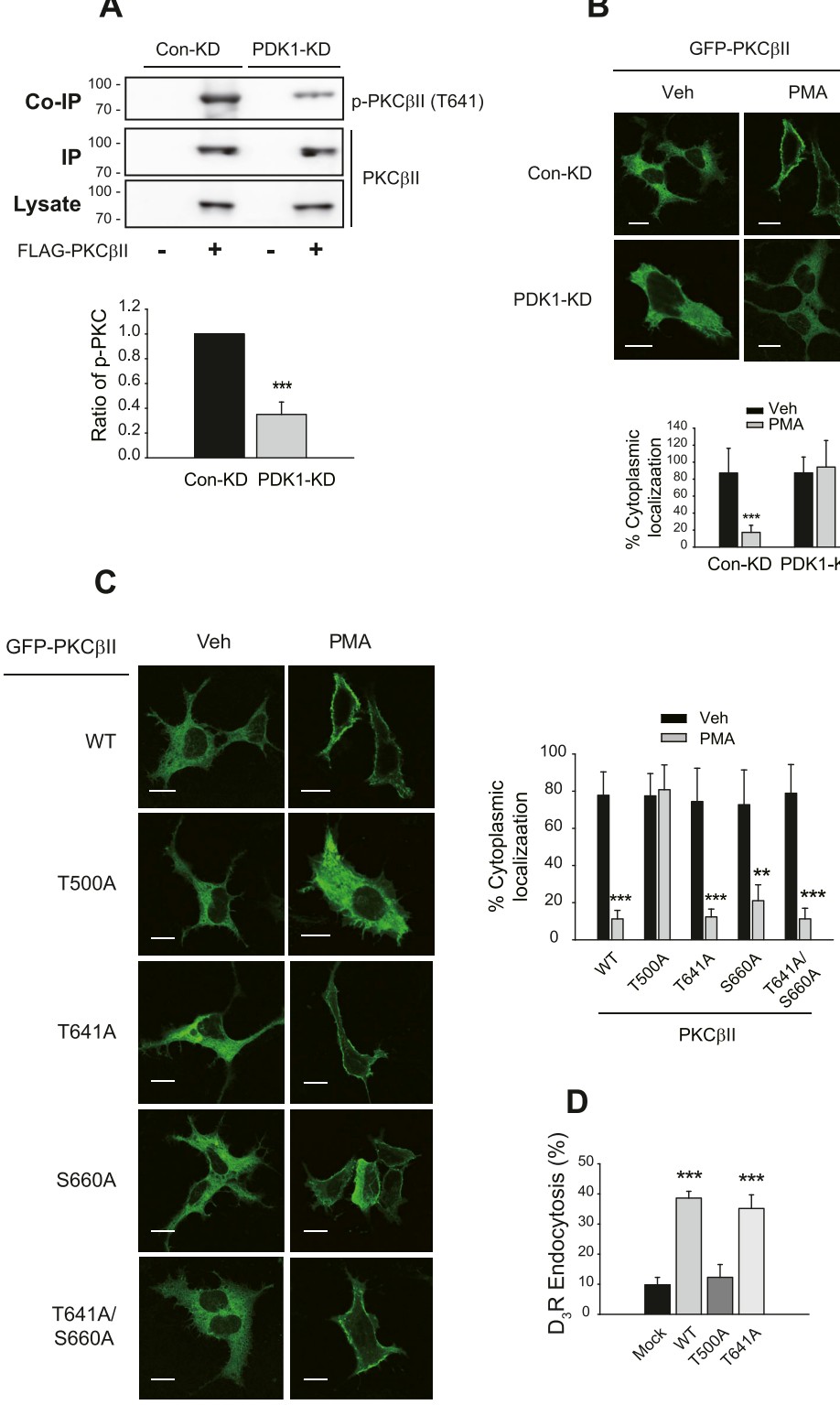

**Figure 4. PDK1-mediated phosphorylation of PKCβII at T500 is required for the PMA-induced translocation to the plasma membrane.**

**(A)** Con-KD and PDK1-KD HEK-293 cells were transfected with FLAG-PKCβII. Cell lysates were immunoprecipitated with anti-FLAG agarose beads. Co-IP and lysates/IPs were immunoblotted with antibodies against phosphor-PKCα/βII (T638/641) and FLAG, respectively. ***$P < 0.001$ compared with the Con-KD group (n = 3). **(B)** Con-KD and PDK1-KD HEK-293 cells were transfected with GFP-PKCβII. Cells were treated with vehicle or 100 nM PMA for 15 min. ***$P < 0.001$ compared with other groups (n = 5). Horizontal bars represent 10 $\mu m$. **(C)** HEK-293 cells were transfected with GFP-tagged WT, T500A, T641A, S660A, or T641A/S660A-PKCβII. Cells were treated with vehicle or 100 nM PMA for 15 min. **$P < 0.01$, ***$P < 0.001$ compared with each vehicle group (n = 5). Horizontal bars represent 10 $\mu m$. **(D)** HEK-293 cells expressing $D_3R$ were transfected with mock, WT-, T500A-, or T641A-PKCβII. Cells were treated with vehicle or PMA 100 nM for 30 min. ***$P < 0.001$ compared with the mock or T500A-PKCβII groups (n = 3).

prerequisite for the PMA-induced nuclear entry, which is required for Mdm2-mediated ubiquitination. Ubiquitination in turn determines the inducible phosphorylation of PKCβII at T500 and T641. Thus, we determined how the inducible phosphorylation of PKCβII

at T500 and T641 is functionally connected to the interaction with actin.

As reported previously (Ikenoue et al, 2008), PKCβII interacted with rictor, a selective component of mTORC2, in response to PMA

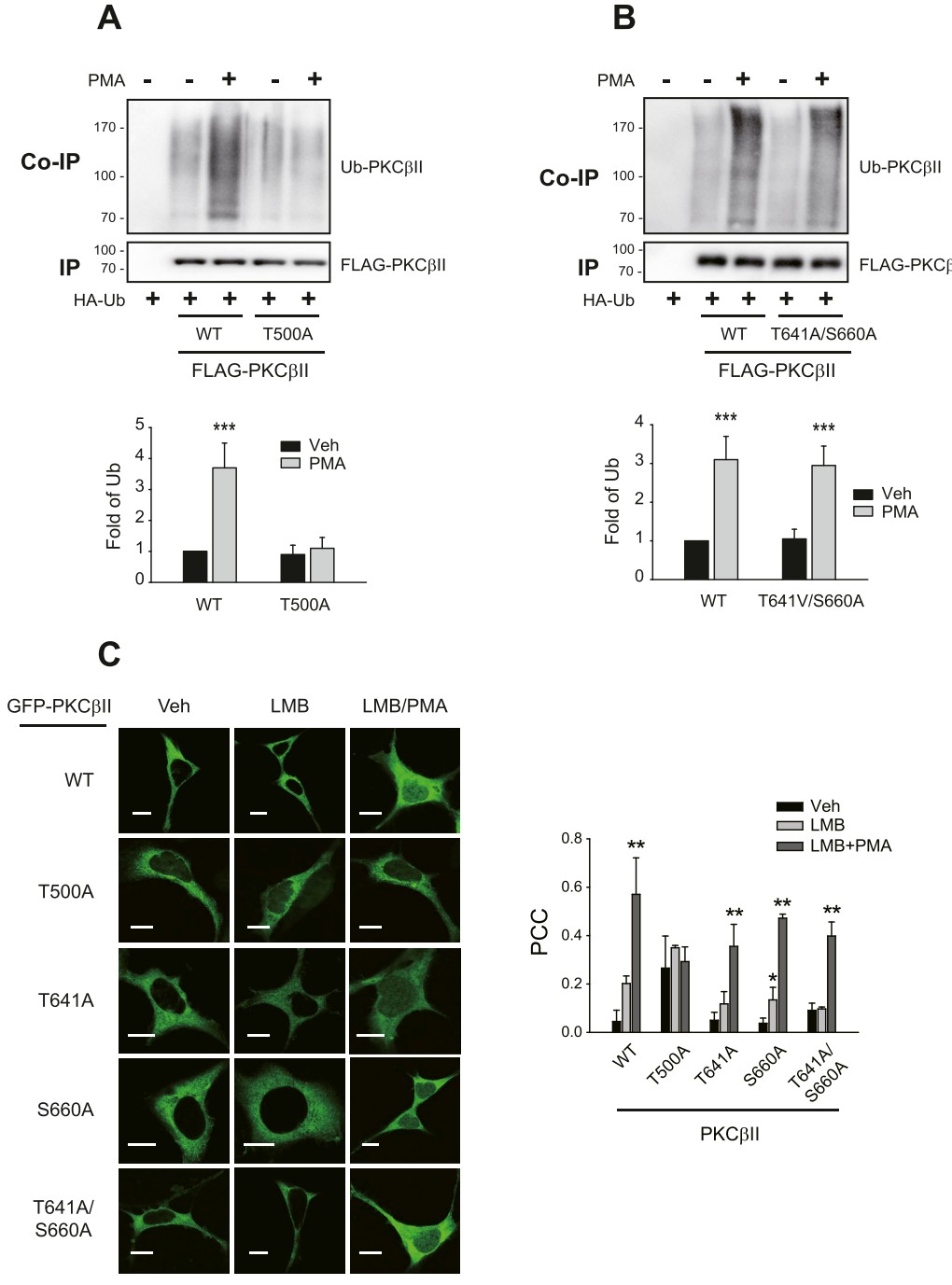

**Figure 5. PDK1-mediated PKCβII phosphorylation at T500 is needed for the nuclear entry and ubiquitination of PKCβII.**
**(A)** HEK-293 cells were transfected with HA-Ub and FLAG-tagged WT- or T500A-PKCβII. The cells were treated with 100 nM PMA for 15 min. Cell lysates were immunoprecipitated with anti-FLAG agarose beads. Co-IPs and IPs were immunoblotted with antibodies against HA and FLAG, respectively. **(B)** HEK-293 cells were transfected with HA-Ub and FLAG-tagged WT- or T641A/S660A-PKCβII. The cells were treated with 100 nM PMA for 15 min. Cell lysates were immunoprecipitated with anti-FLAG agarose beads. Co-IPs and IPs were immunoblotted with antibodies against HA and FLAG, respectively. **(C)** HEK-293 cells were transfected with GFP-tagged WT-, T500A-, T641A-, S660A-, or T641A/S660A-PKCβII. Cells were pretreated with vehicle or 30 ng/ml leptomycin B (LMB) for 6 h, followed by treatment with vehicle or 100 nM PMA for 15 min. PCC represents Pearson's correlation coefficient. *P < 0.05, **P < 0.01 compared with each vehicle group (n = 5). Horizontal bars represent 10 μm.

treatment in our experiments (Fig S5A). Pretreatment with torin1, an mTORC2 inhibitor, blocked T641 phosphorylation but did not affect T500 phosphorylation (Fig S5B). T500A-PKCβII failed to interact with actin (Fig S5C) and rictor (Fig S5D). These results overall suggest that the phosphorylation of PKCβII at T641 is mediated by mTORC2 and that T500 phosphorylation is a prerequisite for the interaction with actin and mTORC2-mediated phosphorylation of PKCβII at T641.

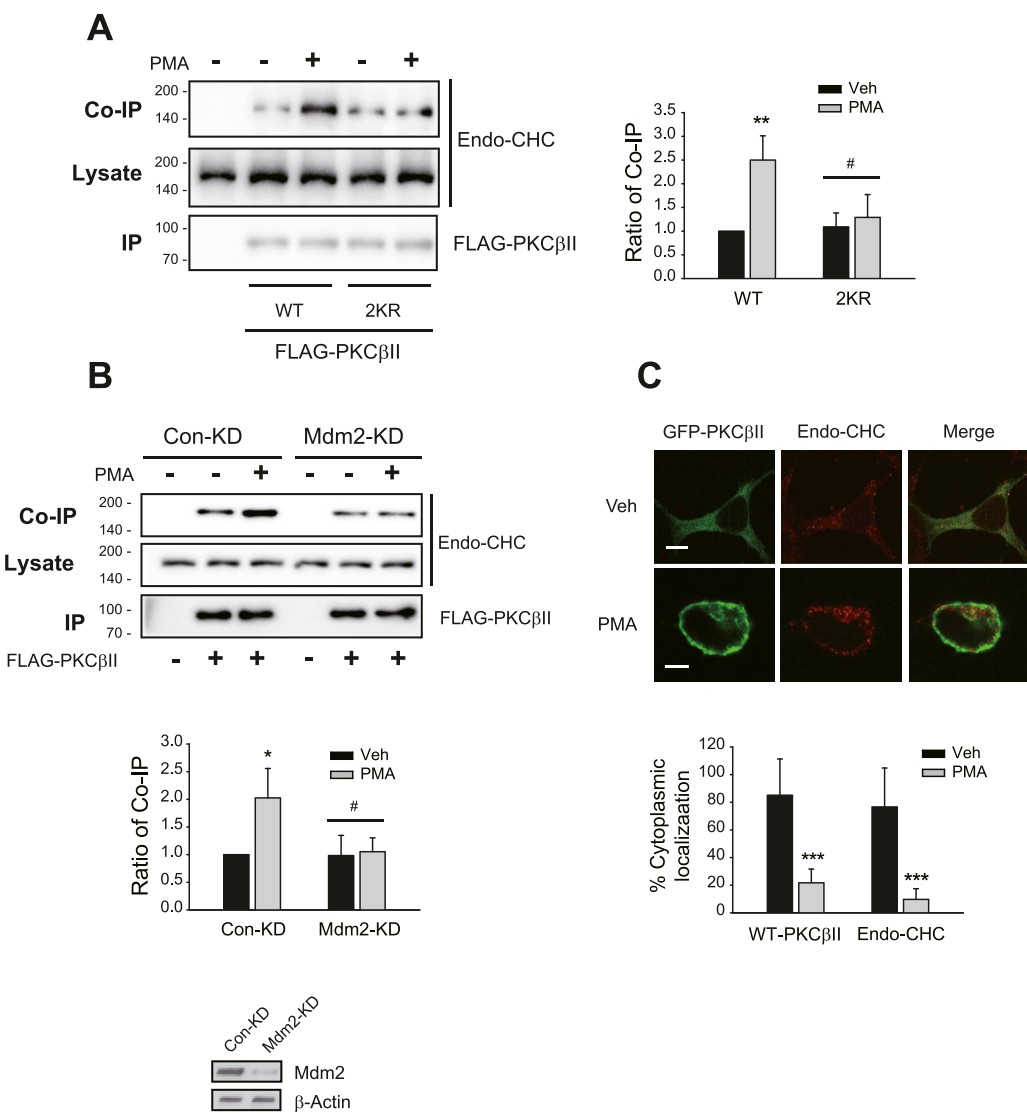

**Figure 6. Mdm2-mediated ubiquitination of PKCβII is needed for the interaction with clathrin.**
**(A)** HEK-293 cells were transfected with FLAG-tagged WT- or 2KR-PKCβII. Cells were treated with 100 nM PMA for 5 min. Cell lysates were immunoprecipitated with anti-FLAG agarose beads. Co-IP/lysates and IPs were immunoblotted with antibodies against clathrin and FLAG, respectively. **$P < 0.01$ compared with the vehicle group, #$P <$ 0.05 compared with the PMA/WT-PKCβII group (n = 3). **(B)** Con-KD and Mdm2-KD HEK-293 were transfected with FLAG-tagged WT- or 2KR-PKCβII. Cells were treated with 100 nM PMA for 5 min. Cell lysates were immunoprecipitated with anti-FLAG agarose beads. Co-IP/lysates and IPs were immunoblotted with antibodies against clathrin and FLAG, respectively. *$P < 0.05$ compared with the vehicle group; #$P < 0.05$ compared with the PMA/Con-KD group (n = 3). Knockdown efficiency for Mdm2 was about 90%. **(C)** HEK-293 cells were transfected with GFP-PKCβII. Cells were treated with vehicle or 100 nM PMA for 15 min. Cells were labeled with anti-clathrin antibodies (1:1,000), followed by Alexa 555–conjugated secondary antibodies (1:500). ***$P < 0.001$ compared with each vehicle group (n = 5). Horizontal bars represent 10 $\mu$m.

Next, we examined whether PKCβII phosphorylation at T641 occurs before or after it binds with actin. As shown in Fig 9A, the phosphorylation of ActX-PKCβII increased at T500 but not at T641 after PMA treatment, suggesting that interaction with actin is needed for the T641 phosphorylation to occur. In addition, ActX-PKCβII did not dissociate from PDK1 after PMA treatment (Fig 9B), suggesting that the interaction with actin is needed for the dissociation of PDK1 from PKCβII, which is a prerequisite for the phosphorylation of PKCβII at T641. In agreement with these results, WT-PKCβII but not ActX-PKCβII interacted with rictor (Fig 9C).

Finally, we determined whether the interaction with actin occurs in the nucleus or after PKCβII moves out of the nucleus.

As shown in Fig 9D, PMA treatment induced the association between PKCβII and actin, and pretreatment with LMB prevented it.

These results suggest that ubiquitinated PKCβII, which can be inducibly phosphorylated at T500 (Fig 8A), moves out of the nucleus and interacts with actin (Fig 9D), dissociates from PDK1 (Fig 8B), and interacts with mTORC2 to get phosphorylated at T641 (Fig 9C). It has been reported earlier that binding with actin increases the autophosphorylation of PKCβII (Blobe et al, 1996). This finding is in agreement with our study, which also shows that actin binding is required for the phosphorylation of PKCβII at T641.

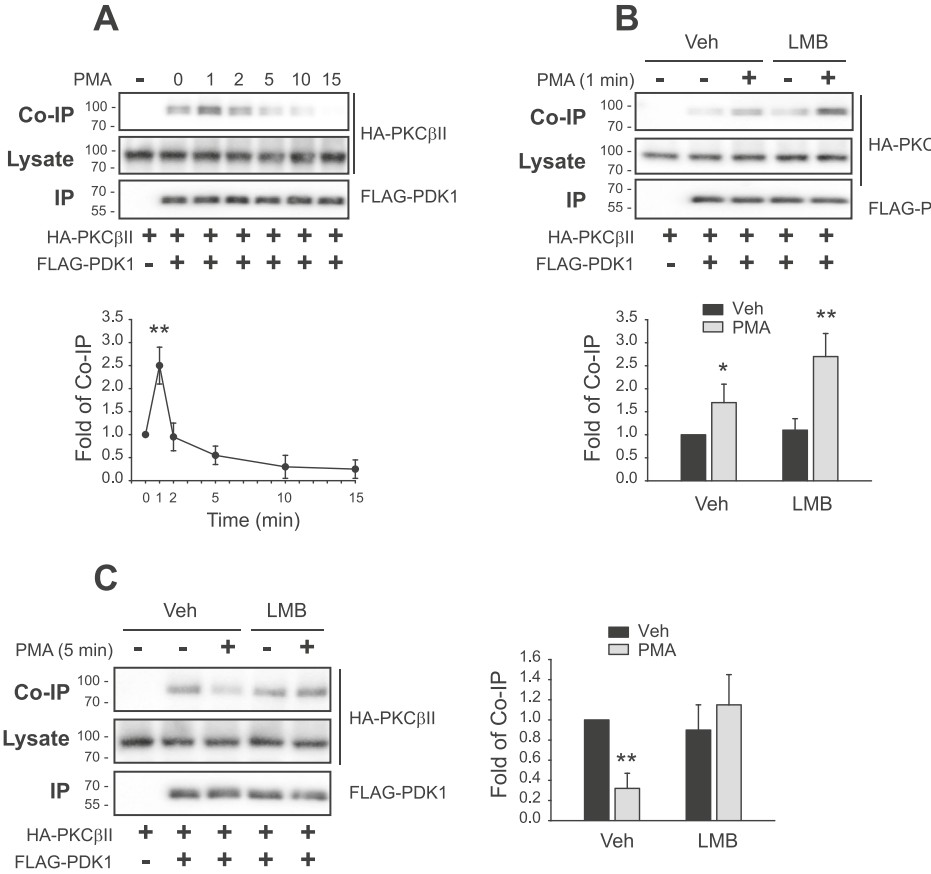

**Figure 7. PMA-induced interaction between PDK1 and PKCβII occurs in the nucleus and is followed by nuclear export and dissociation in the cytosol.**
HEK-293 cells were transfected with HA-PKCβII and FLAG-PDK1. Cell lysates were immunoprecipitated with anti-FLAG agarose beads. Co-IP/lysates and IPs were immunoblotted with antibodies against HA and FLAG, respectively. **(A)** Cells were treated with 100 nM PMA for 0–15 min. **$P < 0.05$ compared with the 0-min group (n = 3). **(B)** Cells were pretreated with vehicle or 30 ng/ml LMB for 6 h, then with 100 nM PMA for 5 min. *$P < 0.05$, **$P < 0.01$ compared with each vehicle (Veh) group (n = 3). **(C)** Cells were pretreated with vehicle or 30 ng/ml LMB for 6 h, then with 100 nM PMA for 5 min. **$P < 0.01$ compared with each Veh/Veh group (n = 3).

## PKCβII interacts with clathrin after it is phosphorylated at T641 accompanied by interaction with actin

As for actin, we determined whether PKCβII interacts with clathrin in the nucleus or cytosol. We found that pretreatment with LMB blocked the PMA-induced interaction between PKCβII and clathrin (Fig 10A), suggesting that PKCβII interacted with actin after PKCβII moved out of the nucleus. CHCX-PKCβII was phosphorylated both at T500 and T641 after PMA treatment (Fig 10B), which indicates that binding to clathrin is not required for the phosphorylation of PKCβII at T500 and T641, that is, the interaction with clathrin occurs after both sites (T500 and T641) are phosphorylated. In addition, both WT-PKCβII and CHCX-PKCβII interacted with rictor (Fig 10C), suggesting that binding to clathrin occurs after mTORC2 binds to PKCβII at T641.

## Discussion

The currently established mechanism of conventional PKC family activation entails translocation to the plasma membrane and conformational changes. Shortly after being synthesized, PKCβII, a member of the conventional PKC family, is phosphorylated at three major sites (A-loop, TM, HM) and stays in the cytosol in the "primed" state. Even though PKCβII is inactive in the primed state because

the pseudosubstrate is still bound to the substrate binding site, it retains its competency for activation by cellular stimuli (Freeley et al, 2011; Newton, 2018). With the rise of intracellular Ca$^{2+}$ levels, PKCβII is recruited to underneath the plasma membrane where DAG and phosphatidylserine bind to the C1 domain. Thereafter, PKCβII undergoes conformational changes, accompanied by the dissociation of pseudosubstrate from the C4 domain, leading to the catalytic activation of PKCβII (Nakamura & Nishizuka, 1994; Newton, 1995).

Current studies are showing that the phosphorylation of PKCβII at three major motifs plays critical roles in its activation (Orr & Newton, 1994; Edwards & Newton, 1997; Edwards et al, 1999). The phosphorylation of PKCβII is known to occur in a constitutive or inducible manner. Along with phosphorylation, Mdm2-mediated ubiquitination of PKCβII is known to be responsible for the activation of PKCβII (Min et al, 2019). And our study showed that it was K63-linked polyubiquitination, suggesting that Mdm2-mediated PKCβII ubiquitination may be involved in the regulation of cellular signaling (Chau et al, 1989). However, more information is needed for a clearer understanding of the specific roles of phosphorylation and ubiquitination in PKCβII activation and how two processes are regulated.

In this study, we determined the molecular mechanisms involved in the activation of PKCβII based on its phosphorylation at three conserved phosphorylation sites; its Mdm2-mediated ubiquitination; and its interactions with Mdm2, actin, and clathrin.

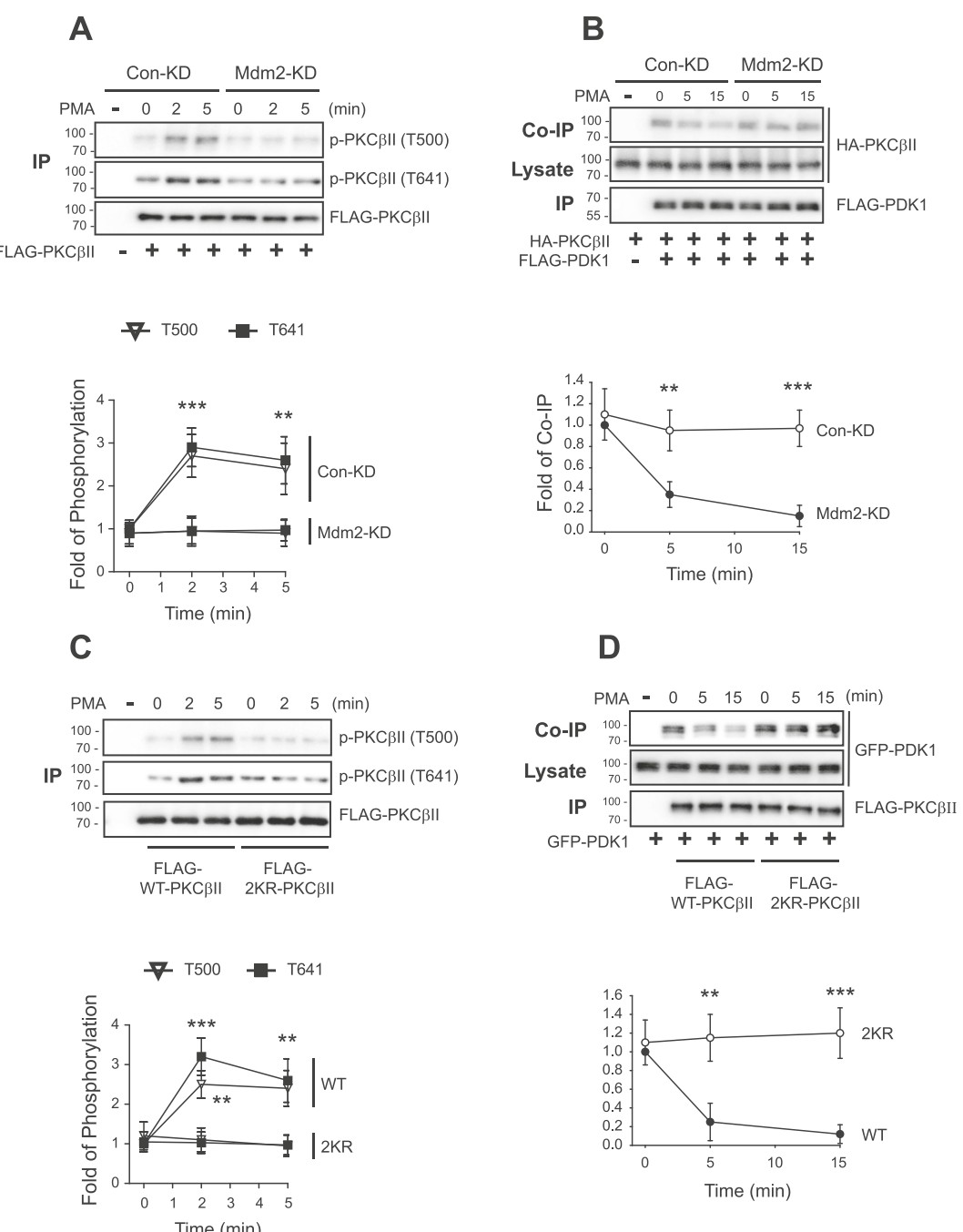

**Figure 8. Mdm2-mediated ubiquitination of PKCβII is needed for the PDK1-mediated inducible phosphorylation of PKCβII at T500.**
**(A)** Con-KD and Mdm2-KD HEK-293 cells were transfected with FLAG-PKCβII. The cells were treated with 100 nM PMA for 0–5 min. IPs were immunoblotted with antibodies against PKCβII phosphorylated at T500 or T641 or antibodies against FLAG. **P < 0.01, ***P < 0.001 compared with the Mdm2-KD group (time point by time point) (n = 3).
**(B)** Con-KD and Mdm2-KD HEK-293 cells were transfected with HA-PKCβII and FLAG-PDK1. The cells were treated with 100 nM PMA for 0–15 min. Cell lysates were immunoprecipitated with anti-FLAG agarose beads. Co-IP/lysates and IPs were immunoblotted with antibodies against HA and FLAG, respectively. **P < 0.01, ***P < 0.001 compared with the Mdm2-KD group (time point by time point) (n = 3). **(C)** HEK-293 cells were transfected with FLAG-tagged WT- or 2KR-PKCβII. The cells were treated with 100 nM PMA for 0–5 min. Cell lysates were immunoprecipitated with anti-FLAG agarose beads. IPs were immunoblotted with antibodies against PKCβII phosphorylated at T500 or T641 or antibodies against FLAG. **P < 0.01, ***P < 0.001 compared with the 2KR-PKCβII group (time point by time point) (n = 3). **(D)** HEK-293 cells were transfected with HA-PKCβII and FLAG-PDK1. The cells were treated with 100 nM PMA for 0–15 min. Cell lysates were immunoprecipitated with anti-FLAG agarose beads. Co-IP/lysates and IPs were immunoblotted with antibodies against HA and FLAG. **P < 0.01, ***P < 0.001 compared with the WT-PKCβII group (time point by time point) (n = 3).

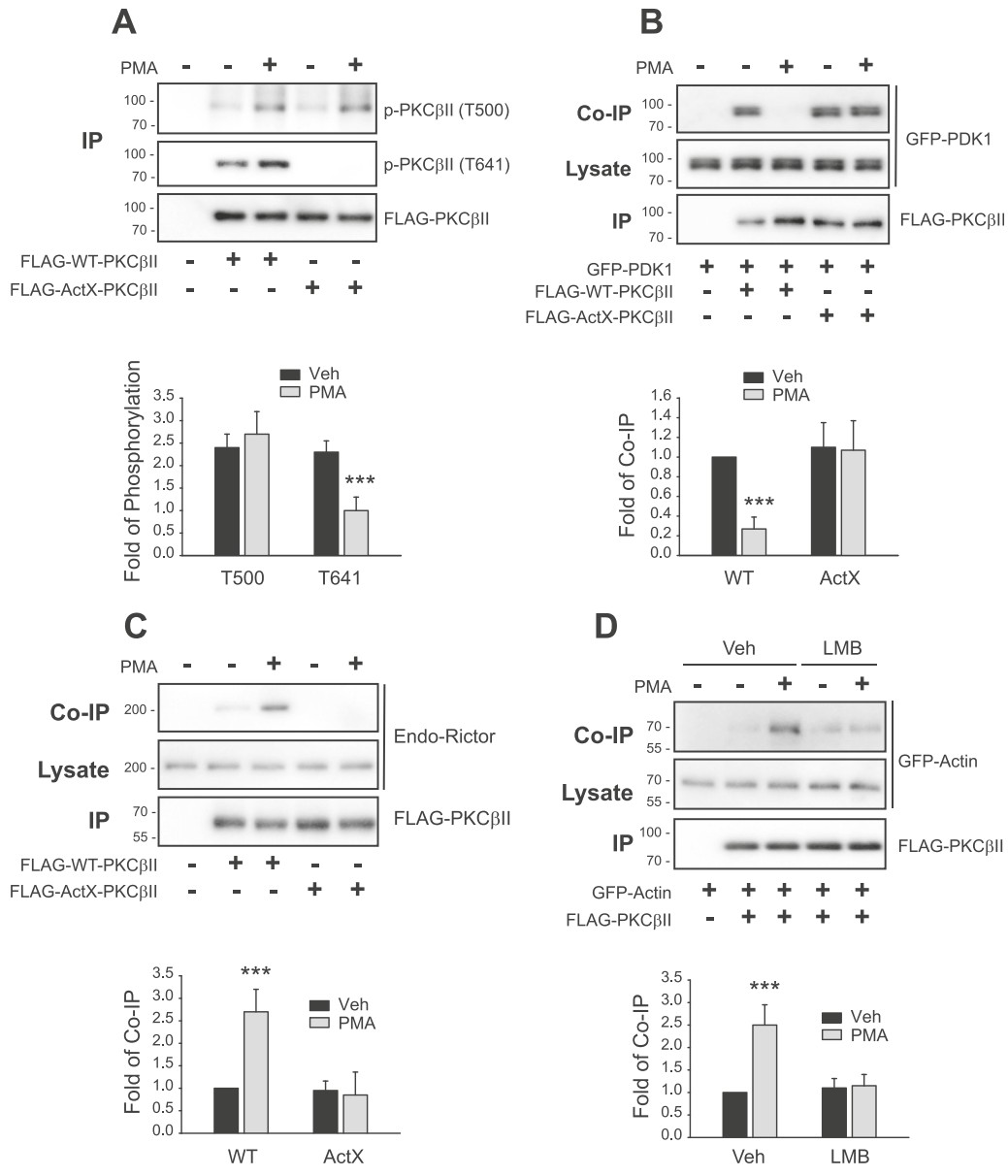

**Figure 9.   Binding with actin is needed for the PMA-induced phosphorylation of PKCβII at T641 and interaction with mTORC2.**
Cells were treated with 100 nM PMA for 5 min, and the cell lysates were immunoprecipitated with anti-FLAG agarose beads. **(A)** HEK-293 cells were transfected with FLAG-tagged WT- or ActX-PKCβII. IPs were immunoblotted with antibodies against PKCβII phosphorylated at T500 or T641 or antibodies against FLAG. ***P < 0.001 compared with the Veh group (n = 3). **(B)** HEK-293 cells were transfected with GFP-PDK1 together with FLAG-tagged WT- or ActX-PKCβII. Co-IP/lysates and IPs were immunoblotted with antibodies against GFP and FLAG, respectively. ***P < 0.001 compared with the Veh group (n = 3). **(C)** HEK-293 cells were transfected with FLAG-tagged WT- or ActX-PKCβII. Co-IP/lysates and IPs were immunoblotted with antibodies against rictor and FLAG, respectively. ***P < 0.001 compared with the Veh group (n = 3). **(D)** HEK-293 cells were transfected with GFP-actin and FLAG-PKCβII. Cells were pretreated with vehicle or 30 ng/ml LMB for 6 h, then with 100 nM PMA for 5 min. ***P < 0.001 compared with the Veh group (n = 3).

Fig S6 shows a summary of the current study. Primed PKCβII, which is in an inactive state and constitutively phosphorylated (Pc) at the A-loop, TM, and HM (not shown), enters the nucleus in response to PMA treatment. In the nucleus, PKCβII undergoes Mdm2-mediated ubiquitination, and ubiquitinated PKCβII is in addition phosphorylated at the A-loop by PDK1 (inducible phosphorylation, Pi) and moves out the nucleus. In the cytosol, PKCβII binds with actin in a ubiquitination-dependent manner, detaches from PDK1, and is phosphorylated at T641 by mTORC2.

Thereafter, PKCβII binds to clathrin and translocates to the plasma membrane.

According to the results obtained in this study, the role of phosphorylation differs depending on whether it occurs in a constitutive or an inducible manner. For PKC activation, PKC needs to enter the nucleus to be ubiquitinated by Mdm2 and inducibly phosphorylated by PDK1 in the A-loop. Actin and clathrin are needed for the subsequent phosphorylation at TM and the translocation to the plasma membrane, respectively.

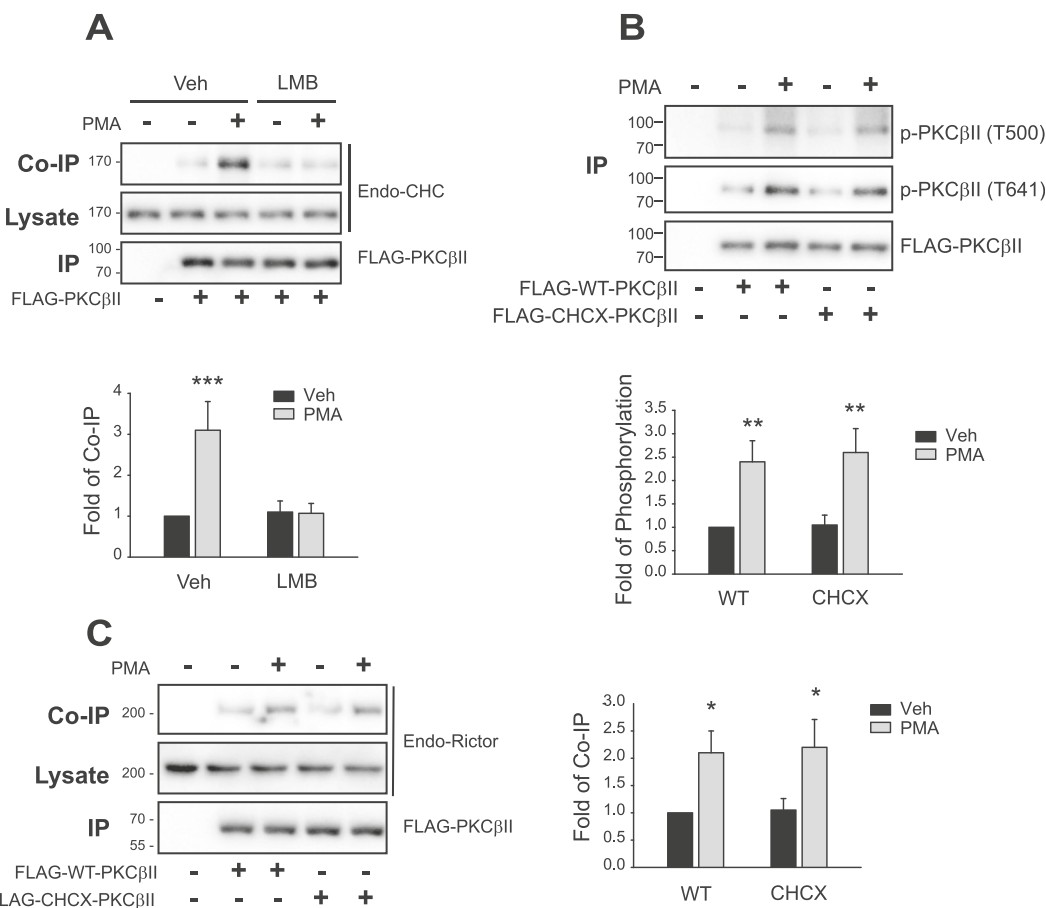

**Figure 10. Phosphorylation of PKCβII at T500 and T641 is needed for the interaction with clathrin.**
Cells were treated with 100 nM PMA for 5 min, and the cell lysates were immunoprecipitated with anti-FLAG agarose beads. **(A)** HEK-293 cells transfected with FLAG-PKCβII were pretreated with vehicle or 30 ng/ml LMB for 6 h, then with 100 nM PMA for 5 min. Co-IP/lysates and IPs were immunoblotted with antibodies against clathrin heavy chain and FLAG, respectively. **(B)** HEK-293 cells were transfected with FLAG-tagged WT- or CHCX-PKCβII. IPs were immunoblotted with antibodies against PKCβII phosphorylated at T500 or T641 or antibodies against FLAG. **P < 0.001 compared with each Veh group (n = 3). **(C)** HEK-293 cells were transfected with FLAG-tagged WT- or CHCX-PKCβII. Co-IP/lysates and IPs were immunoblotted with antibodies against rictor and FLAG, respectively. *P < 0.05 compared with each Veh group (n = 3).

Our study shows that phosphorylation of PKCβII at T500 and T641 occurs in two different ways—constitutive and induced—and that they have different functional meanings. Constitutive phosphorylation of PKCβII at T500, which occurs in the cytosol in a basal state, is required for the nuclear entry of PKCβII in response to PMA treatment and the subsequent ubiquitination in the nucleus. In contrast, inducible phosphorylation occurs after PKCβII enters the nucleus and plays a key role in mediating subsequent PKCβII activation processes that include binding with actin, phosphorylation at T641 by mTORC2, and translocation to the plasma membrane after binding with clathrin.

The fact that constitutive phosphorylation is required for ubiquitination and that, in turn, ubiquitination is necessary for inducible phosphorylation is demonstrated by several experimental results of the current study. For example, PMA treatment increased the interactions between PDK1 and PKCβII at 1–2 min after PMA treatment and the phosphorylation of PKCβII at T500, and both were abolished by mutations in the ubiquitination sites (2KR-PKCβII) or by knockdown of cellular Mdm2 (Fig 8A–C). The basal interaction between PDK1 and PKCβII, on the other hand, was not altered by knockdown of Mdm2 (Fig 8B) or mutation of ubiquitination sites in the PKCβII, 2KR-PKCβII (Fig 8D).

Various functional interactions between actin filament and CME have been reported. For example, the interaction between actin microfilaments and clathrin-coated structures (Salisbury et al, 1980) and requirement of F-actin dynamics at multiple stages of clathrin-coated vesicle formation have been demonstrated (Yarar et al, 2005). Our study suggests another possible role of actin in CME, which includes actin-mediated regulation of the interaction between PKCβII and clathrin.

In conclusion, our study reveals a sequence of cascades involved in PMA-induced PKCβII activation. (1) PDK1-mediated constitutive phosphorylation of PKCβII at T500 in the cytosol plays permissive roles in PKCβII activation by allowing the nuclear entry of PKCβII in response to PMA treatment. (2) PKCβII undergoes K63-linked Mdm2-mediated ubiquitination in the nucleus, which is needed for the interaction with PDK1, resulting in the inducible phosphorylation of PKCβII at T500. (3) Inducible phosphorylation of PKCβII at T500 allows the nuclear export of the ubiquitinated PKCβII and interactions with actin in the cytosol. (4) Interactions with actin lead to

the dissociation of ubiquitinated PKCβII from PDK1, which is needed for the mTORC2-mediated phosphorylation of PKCβII at T641. (5) Ubiquitination and inducible phosphorylation of PKCβII at T641 allow the interaction with clathrin and the translocation to the plasma membrane.

# Materials and Methods

### Reagents

PMA, leptomycin A (LMB), agarose beads coated with monoclonal antibodies against FLAG epitope, rabbit anti-FLAG M2 antibodies (AB_439687), rabbit antibodies against GFP (AB_439690), and HA antibodies (AB_2610070) were purchased from Sigma-Aldrich Chemical Co. Latrunculin A (LatA) was obtained from Cayman Chemical. Mouse monoclonal antibodies against CHC (AB_397865) were obtained from BD Bioscience. Rabbit antibodies against rictor (AB_2179961), phosphor-PKCα/βII (Thr638/641) (AB_2284224), K48-linkage specific polyubiquitin (AB_10557239), and K63-linkage specific polyubiquitin rabbit mAb (HRP conjugate) antibody (AB_2798064) were purchased from Cell Signaling Technology. Anti-rabbit HRP-conjugated secondary antibodies (AB_2533967) and phospho-PKCβ1&2 (Thr500) (AB_2533805) were obtained from Thermo Fisher Scientific, and anti-mouse HRP-conjugated secondary antibodies (AB_10015289) were purchased from Jackson ImmunoResearch Laboratories, Inc. Mdm2 E3 ligase inhibitor and antibodies against Mdm2 (AB_627920) were obtained from Santa Cruz Biotechnology and torin1 from Selleck Chemicals. Alexa Fluor 594–conjugated anti-rabbit (AB_142057) and anti-mouse (AB_141593) antibodies were from Molecular Probes. [3H]-sulpiride was purchased from Perkin Elmer Life Sciences.

### Plasmid constructs

Human PKCβI was obtained from Addgene. PKCβII constructs (FLAG-, HA-, or GFP-tagged) and PKCβII mutant constructs (T500A-, T641A-, S660A-, T641A/S660A-, CHCX-, 2KR-, and ActX-PKCβII) were previously described (Dutil et al, 1994; Edwards et al, 1999; Min et al, 2019) or were prepared via site-directed mutagenesis. HA-Ub, PDK1, and the human D$_3$R were described previously (Guo et al, 2015; Zhang et al, 2020a, 2020b), as were the small hairpin RNAs (shRNAs) of CHC, caveolin1 (Cav1), PDK1, and Mdm2 (Guo et al, 2015; Zhang et al, 2020b).

### Cell culture

Human embryonic kidney 293 (HEK-293, CVCL_0045) cells obtained from the American Type Culture Collection were cultured (37°C, 5% CO$_2$) in minimal essential medium supplemented with 10% fetal bovine serum, 100 U/ml penicillin, and 100 μg/ml streptomycin (Thermo Fisher Scientific). Transfections were performed using polyethylenimine (Polyscience). CHC-KD, Cav1-KD, importin β1-KD, PDK1-KD, and Mdm2-KD cells were prepared by stably expressing shRNAs in PLKO.1 (Sigma-Aldrich Chemical Co.) targeting each gene under puromycin selection. Con-KD cells were prepared by stably

transfecting scrambled shRNAs of the corresponding vectors. Only HEK-293 cells with passage numbers ranging from 35 to 45 were used throughout this study.

### Immunoprecipitation

For immunoprecipitation, cells expressing FLAG-tagged proteins were lysed (4°C, 1 h, rotating) with lysis buffer (20 mM Hepes, 150 mM NaCl, 2 mM EDTA, 10% glycerol, 0.5% Nonidet P-40, 5 μg/ml aprotinin, 5 μg/ml leupeptin, 20 μg/ml phenylmethylsulfonylfluoride, 10 mM NaF, and 1 mM sodium orthovanadate). The cell lysates were then incubated with the agarose beads coated with FLAG antibodies for 2–3 h at 4°C. The beads were washed three times with ice-cold washing buffer (50 mM Tris, pH 7.4, 137 mM NaCl, 10% glycerol, and 1% NP-40) for 5 min each time. The cell lysates and immunoprecipitates (IPs) were analyzed on SDS–PAGE gels and transferred to nitrocellulose membranes (Sigma-Aldrich Chemical Co.). The membranes were incubated with primary antibodies for target protein at 4°C overnight, followed by incubation with HRP-conjugated secondary antibodies. The target proteins were visualized by a chemiluminescent substrate (Thermo Fisher Scientific). Immunoblots were quantified by gray densitometry using the Multi Gauge V3.0 (Fuji Film).

### Immunocytochemistry

Cells were transfected with corresponding cDNA constructs and cultured on glass coverslips. After 24 h, cells were fixed with 4% paraformaldehyde in PBS for 15 min at 25°C and then permeabilized with 0.1% Triton X-100 in PBS for 1 min at 25°C. Cells were blocked with PBS containing 3% FBS and 1% BSA for 1 h and then incubated with corresponding antibodies for 2 h at 25°C. After washing the cells three times with PBS, they were incubated with Alexa 555–conjugated secondary antibodies (1:500) and visualized with a laser-scanning confocal microscope (TCS SP5/AOBS/Tandem). ~5–10 cells were analyzed for each sample, and the same experiments were repeated three times. Images were processed using Image J software, and colocalization was analyzed based on Pearson's correlation coefficient (γ value).

### Detection of ubiquitinated PKCβII

FLAG-PKCβII and HA-Ub were co-transfected into HEK-293 cells. The cells were serum-starved for 4–6 h and then treated with 100 nM PMA for 15 min (Min et al, 2019). Cell lysates were solubilized in a lysis buffer (150 mM NaCl, 50 mM Tris, pH 7.4, 1 mM EDTA, 1% Triton X-100, 10% glycerol, 1 mM sodium orthovanadate, 5 μg/ml leupeptin, 5 μg/ml aprotinin, and 10 mM N-ethylmaleimide) and immunoprecipitated with agarose beads coated with anti-FLAG antibodies. The co-IPs and IPs were analyzed using SDS–PAGE and blotted with antibodies against HA and FLAG, respectively.

### Receptor endocytosis assay

Endocytosis of D$_3$R was determined based on the hydrophilic properties of [3H]-sulpiride (Kim et al, 2001; Guo et al, 2015). HEK-293 cells expressing D$_3$R were seeded (1.5 × 10$^5$ cells/well) 1 d after

transfection on 24-well plates. The following day, the cells were rinsed once and pre-incubated for 15 min with 0.5 ml pre-warmed serum-free medium containing 10 mM Hepes (pH 7.4) at 37°C. The cells were stimulated with 100 nM PMA or 1 $\mu$M isoproterenol for 30 min. They were then incubated with 250 $\mu$l of [$^3$H]-sulpiride (7.2 nM) at 4°C for 150 min in the presence or absence of unlabeled competitive inhibitor (10 $\mu$M haloperidol). The cells were washed thrice with the same medium, and 1% SDS was added. Samples were mixed with 2 ml Lefko-Fluor scintillation fluid and counted on a liquid scintillation analyzer.

## Statistics

All data are expressed as the mean ± SD. Statistical significance was analyzed using paired two-tailed $t$ tests to compare two-group or one-way ANOVA with Tukey's post hoc test for multiple group comparisons. A $P$-value < 0.05 was considered significant.

# Data Availability

Data and the materials used in this study are available to any qualified researcher upon reasonable request addressed to K-M Kim.

# Supplementary Information

# Acknowledgements

The authors acknowledge Dr. Marc G Caron (Duke University) for providing reagents and technical support. We thank the Korea Basic Science Institute for their technical support. This research was supported by Basic Science Research Program through the National Research Foundation of Korea (NRF) funded by the Ministry of Education (KRF-2020R1I1A3062151); Korea Drug Development Fund funded by Ministry of Science and ICT, Ministry of Trade, Industry, and Energy, and Ministry of Health and Welfare (HN21C1076).

## Author Contributions

X Min: resources, data curation, formal analysis, investigation, and methodology.
S Wang: conceptualization, resources, data curation, investigation, and methodology.
X Zhang: resources and investigation.
N Sun: data curation, investigation, and methodology.
K-M Kim: conceptualization, supervision, funding acquisition, and writing—original draft, review, and editing.

## Conflict of Interest Statement

The authors declare that they have no conflict of interest.

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
