## [Reviewer comments · Life Science Alliance]

Life Science Alliance

PKC β II activation requires nuclear trafficking for phosphorylation and Mdm2-mediated ubiquitination

Xiao Min, Shujie Wang, Xiaohan Zhang, Ningning Sun and Kyeong-Man Kim
DOI: <https://doi.org/10.26508/lsa.202201748>

Corresponding author(s): Prof. Kyeong-Man Kim (Chonnam National University)

Review Timeline:

Submission Date:	2022-09-29
Editorial Decision:	2022-10-27
Revision Received:	2022-12-28
Editorial Decision:	2023-01-09
Revision Received:	2023-01-11
Accepted:	2023-01-11

Transaction Report:

October 27, 2022

Re: Life Science Alliance manuscript #LSA-2022-01748-T

Prof. Kyeong-Man Kim
Chonnam National University
Pharmacology, College of Pharmacy
300 Yong Bong Dong
Gwangju 500-757
Korea, Republic of

Dear Dr. Kim,

Thank you for submitting your manuscript entitled "PKC β II activation requires nuclear trafficking for phosphorylation and Mdm2-mediated ubiquitination" to Life Science Alliance. The manuscript was assessed by expert reviewers, whose comments are appended to this letter. We invite you to submit a revised manuscript addressing the Reviewer comments.

Thank you for this interesting contribution to Life Science Alliance. We are looking forward to receiving your revised manuscript.

Sincerely,

B. MANUSCRIPT ORGANIZATION AND FORMATTING:

Reviewer #1 (Comments to the Authors (Required)):

PKC- β II induces endocytosis of GPCRs. This study is interesting, as it provides a new activation mechanism for PKC- β II involved in the regulation of GPCRs.

Previously, Dr. Kim's group found that ubiquitination is required for PKC- β II activation. This study further advances the activation mechanism of PKC- β II. It reveals that PKC- β II undergoes ubiquitination in the nucleus and that this ubiquitination is required for its translocation to the plasma membrane. There are three types of ubiquitination: mono-, multi-, and poly. The lysine residue of poly-ubiquitination is either at position 48 (K48) or 63 (K63), leading to different functions (such as differentiation and cellular signaling). It would be of interest to clarify which modification form is responsible for PKC- β II cycling.

Reviewer #2 (Comments to the Authors (Required)):

Manuscript Review: PKC II Activation Requires Nuclear Trafficking for Phosphorylation and Mdm2-Mediated Ubiquitination

Summary: In this work, Xiao Min and colleagues investigate step by step mechanisms involved in PKC II activation. They report on regulation of ubiquitination, phosphorylation, and translocation of PKC β II. They propose a pathway by which stimulation of PKC results in its movement to the nucleus, ubiquitination, exit from nucleus, and further phosphorylation prior to association with the plasma membrane. This is a very ambitious proposal that is not fully supported by the presented results. This reviewer would have appreciated a more focused and better 'defended' proposal based on very clear results.

Major Points:

1. The quality of the results is very variable. Some are quite clear and convincing (e.g. 1B, 1C, etc, 9B) while others are marginal (e.g., 3B, 3C, 3D, 4C, etc).
2. Many of the main conclusions are over concluded and based on unclear data (e.g. translocation to the nucleus, ubiquitination, LMB effects, etc).
3. The nuclear localization is over concluded and based primarily on the results in Fig 5C, which are not clear, and on the effects of LMB on co IP of PKC and PDK1, which are modest. Also, it is not clear that LMB is used at optimal concentrations or that its effects are solely due to the purported mechanism of action on exportin. Since the authors are using a concentration of LMB relatively higher than what is needed to inhibit nuclear export, they should do a dose response in the 2-20 nM (4-36 ng/ml) range.
4. The overall approach and scheme totally ignore interactions of PKC with phosphatidylserine or calcium.
5. There is no investigation of mechanisms of nuclear translocation. Also, if pT500 is required for nuclear entry, how come phosphorylation also occurs in the nucleus. What is that mechanism?

Other Points:

1. Abstract. "Our results show that PDK1-mediate constitutive phosphorylation of PKC II at the activation-loop (T500) is required for..." This needs revision in the abstract, it is very confusing. PDK is both required for T500 phosphorylation in the cytosol and T500 phosphorylation following ubiquitination? The abstract could actually use a total re write.
2. Fig 1B. Not clear what those constructs are; are they mutants? What does 'combine' refer to? Is the ActX construct active?
3. Fig 1C. How many cells counted? Same question for similar results with pictures. Figure 1C and S1C: Additionally, the images are not entirely clear and the LatA condition seems more intense (maybe more gain on the software?). The vehicle

+PMA condition morphology looks rather odd compared to the cells presented.

4. Fig 1D and 2D. what is measured? This should be shown on the y axis

5. Figure S1D: Does the label KD cells generalize Cav1-KD and CHC-KD? If so there two snips for Cav1 and CHC are different blots and should each have their own appropriate actin control. This validation is important since there is a claim that the endocytosis is Catherin mediated and independent of caveolin.

6. Fig 2A. The quantitation does not seem to match the figures. What do the black and grey bars refer to? The quantitation is missing a legend. Cav1-KD +PMA should membrane localize, why does the quantitation show high cytosolic presence?

7. Figure 2B and Others: A minor point for the blots. The actin-GFP + row is not necessary, makes the figures convoluted. Also, not sure why actin is being overexpressed in the cells. Expression of actin is endogenously high, and antibody directed against the protein are specific and tend to give clean blots.

8. Fig 3A. The labeling of the figure (and many others) is not clear. What is being IP'd? what antibodies are being used for western blots?

9. Fig. 3 Where are the higher MW bands in the FLAG-PKCbII? This applies to all panels in this figure.

10. Fig 3B. the results are not convincing

11. Fig 3C. results are not convincing.

12. Page 10 bottom. Where is the evidence that ubiquitination is needed (allowing) for translocation?

13. Fig 4A. it appears that PDK1 has a modest effect on pPKC T641 (compared with the total pull down).

14. Figure 5A: Why is there a change in construct used for PKCBII? The previous figures used T500A to assess membrane localization in absence of PDK1 phosphorylation; here PMA treated ubiquitination of PKCBII T500V is assessed instead.

15. Fig 5C. the results are not convincing. What is LMB? What is PCC? Where are results with PMA alone on nuclear localization? What is the Y-Axis and there is no description of this quantitation in the methods or figure captions. Also, it is rather odd that the quantitation of the T500A veh is so high compared to the others and has a large error.

16. Fig 6A. results of the western are not very convincing. Is the 2KR PKC active as a kinase?

17. Fig 6B. interaction seems somewhat reduced but not absent (conclusion p12: "PKC β failed to interact, Δ ")

18. Fig 6C. the middle images are not clear. The PMA treated conditions may look membrane localized at first glance, but it is hard to tell where the nucleus is and where the cytoplasm is located. This appears as a cell with a high nucleo to cytoplasm ratio, making the sliver of space surrounding the circle be the cytosol and near the plasma membrane.

19. Figure S2A: Lane 4-7 for Actin-GFP look very similar in intensity and do not match the quantitation.

20. Figure S2B: Same issue, it is not obvious that the quantitation accurately reflects the actin-GFP co-IP. However, there might be an issue with the image quality that makes it hard to visualize.

21. Fig 7B. that stats should compare the two PMA treatments for differences.

22. Fig 7C. this is more complicated than the text indicates as basal interaction is higher with LMB. For both 7B and C, it may be better to show as a time course similar to A,

23. It would be of high significance to show the phosphorylation of PKC β on T500 and S641 in panels B and C.

24. Fig S1C, D: T500V

25. Fig. 9B and C. Does LatA treatment have the same consequence?

26. Page 11: phosphatidylserine not phosphoserine

Reviewer #1 (Comments to the Authors (Required)):

PKC- β II induces endocytosis of GPCRs. This study is interesting, as it provides a new activation mechanism for PKC- β II involved in the regulation of GPCRs.

Previously, Dr. Kim's group found that ubiquitination is required for PKC- β II activation. This study further advances the activation mechanism of PKC- β II. It reveals that PKC- β II undergoes ubiquitination in the nucleus and that this ubiquitination is required for its translocation to the plasma membrane. There are three types of ubiquitination: mono-, multi-, and poly. The lysine residue of poly-ubiquitination is either at position 48 (K48) or 63 (K63), leading to different functions (such as differentiation and cellular signaling). It would be of interest to clarify which modification form is responsible for PKC- β II cycling.

→ In response to the reviewer's comment, we determined the modification form of PKC β II ubiquitination, which is apparently polyubiquitination on SDS-PAGE. Our results showed that PKC β II undergoes K63-linked polyubiquitination in response to PMA treatment. These results are shown in Fig. S2 and the functional meaning of K63-linked polyubiquitination is described in the discussion.

Reviewer #2 (Comments to the Authors (Required)):

Manuscript Review: PKC β II Activation Requires Nuclear Trafficking for Phosphorylation and Mdm2-Mediated Ubiquitination

Summary: In this work, Xiao Min and colleagues investigate step by step mechanisms involved in PKC β II activation. They report on regulation of ubiquitination, phosphorylation, and translocation of PKC β II. They propose a pathway by which stimulation of PKC results in its movement to the nucleus, ubiquitination, exit from nucleus, and further phosphorylation prior to association with the plasma membrane. This is a very ambitious proposal that is not fully supported by the presented results. This reviewer would have appreciated a more focused and better 'defended' proposal based on very clear results.

Major Points:

1. The quality of the results is very variable. Some are quite clear and convincing (e.g. 1B, 1C, etc, 9B) while others are marginal (e.g., 3B, 3C, 3D, 4C, etc).

1) Figs. 3B -3D. It seems that the reviewer is not particularly convinced about the results of ubiquitination experiments. This is very understandable, because ubiquitin of 8.6 kDa is attached in various numbers, so the target protein that has been ubiquitinated can only be seen diffusely on SDS-PAGE gel. When compared with the previously published experimental results (Lee et al., 2019; Shenoy et al., 2009), we believe that the quality of these experimental results is not inferior to those of previous ones.

2) Figs. 1B, 1C, 9B. When analyzing data, the ratio of co-IP and IP was calculated and statistically processed. In these blots, some inequalities were observed for IPs, but co-IPs showed clear differences, showing statistical significance.

3) Fig. 4C. When the original images used in this figure were compared with those used in other figures, the quality was the same. The reason why the resolution of these figures appears to be low is that the size of the images was kept small due to spatial limitations. In the revised figure, we increased the size of these images as much as possible.

2. Many of the main conclusions are over concluded and based on unclear data (e.g. translocation to the nucleus, ubiquitination, LMB effects, etc).

→ This comment overlaps with comment #3 and we answered this comment along with comment #3.

3. The nuclear localization is over concluded and based primarily on the results in Fig 5C, which are not clear, and on the effects of LMB on co IP of PKC and PDK1, which are modest. Also, it is not clear that LMB is used at optimal concentrations or that its effects are solely due to the purported mechanism of action on exportin. Since the authors are using a concentration of LMB relatively higher than what is needed to inhibit nuclear export, they should do a dose response in the 2-20 nM (4-36 ng/ml) range.

3-1) LMB concentration: It is true that LMB is usually used at 20 nM, which is lower than 30 nM, the concentration used in our study. In previous studies, we have used LMB at concentrations of 20 or 30 nM. Although there was no significant difference at the two concentrations, 30 nM was used in this study to obtain more reliable results. We think that 30 nM is a reasonable concentration of LMB for the inhibition of nuclear export because some studies use 40 nM (Shen et al., 2006; Tsai et al., 2022) or even 200 nM (Engel et al., 1998).

3-2) If we had relied solely on Fig. 5C to assert that PKC β II enters the nucleus and is ubiquitinated by Mdm2, we would agree with the reviewer. However, a previously study clearly demonstrated in a previous publication (Min et al., 2019) that PKC β II enters the nucleus in response to PMA treatment (or stimulation of angiotensin type 1A receptor) and is ubiquitinated by Mdm2 in the nucleus. In this previous study, we showed that PKC β II enters the nucleus by immunocytochemistry and subcellular fractionation. In addition, the relationship between nuclear entry and Mdm2-mediated ubiquitination was demonstrated by utilizing Mdm2 mutants in the nuclear import regulatory region.

3-3) In the case of Fig.5C, the reviewer gave another but similar additional comment (#15), so a more detailed answer is given in query #15.

4. The overall approach and scheme totally ignore interactions of PKC with phosphatidylserine or calcium.

→ We sincerely apologize to the reviewer if the diagram we presented gave the impression of ignoring the facts that have been firmly proven in PKC research. But we, by no means, intended to ignore the role of phosphatidylserine (PS) and calcium in PKC activation. PS is involved in the activation of most PKC isoforms, and calcium is involved in the activation of all conventional PKC isoforms. Thus, we thought it would be better to omit PS and

calcium in the research article which has limited space and rather focus on new discoveries. They can be added together when we write a review paper later.

5. There is no investigation of mechanisms of nuclear translocation. Also, if pT500 is required for nuclear entry, how come phosphorylation also occurs in the nucleus. What is that mechanism?

→ While revising this submission, a paper was published that identified the mechanism of nuclear translocation of PKC β II (Min et al., 2023). The key contents are as follows.

"Phorbol 12-myristate 13-acetate (PMA)-induced phosphorylation and ubiquitination of PKC β II, which are needed for its translocation to the plasma membrane, required the presence of both G β γ and 14-3-3 ϵ . G β γ and 14-3-3 ϵ mediated the constitutive phosphorylation of PKC β II by scaffolding PI3K and PDK1 in the cytosol, which is an inactive but required state for the activation of PKC β II by subsequent signals. In response to PMA treatment, the signaling complex translocated to the nucleus with dissociation of PI3K from it. Thereafter, PDK1 stably interacted with 14-3-3 ϵ and was dephosphorylated; PKC β II interacted with Mdm2 along with G β γ , leading to its ubiquitination at two lysine residues on its C-tail. Finally, PDK1/14-3-3 ϵ and ubiquitinated PKC β II translocated to the plasma membrane."

Other Points:

1. Abstract. "Our results show that PDK1-mediate constitutive phosphorylation of PKC β II at the activation-loop (T500) is required for..." This needs revision in the abstract, it is very confusing. PDK is both required for T500 phosphorylation in the cytosol and T500 phosphorylation following ubiquitination? The abstract could actually use a total re write.

→ It's understandable that reviewers will feel very confused and frustrated at this point. In this regard, we recommend that reviewers read the two review papers first (Freeley et al., 2011; Newton, 2018). Of course, it is very strange how constitutive and induced phosphorylation can occur on the same residue, but it was clearly observed in our study and many existing papers have reported the same results. Although it is an assumption, perhaps constitutive phosphorylation, dephosphorylation, and inducible phosphorylation occur sequentially.

- A summary of the constitutive and inducible phosphorylation of PKC excerpted from the review paper is as follows. "Protein Kinase C (PKC) is a family of serine/threonine kinases whose function is influenced by phosphorylation. In particular, three conserved phosphorylation sites known as the activation-loop, the turn-motif and the hydrophobic-motif play important roles in controlling the catalytic activity, stability and intracellular localisation of the enzyme. Prevailing models of PKC phosphorylation suggest that phosphorylation of these sites occurs shortly following synthesis and that these modifications are required for the processing of newly-transcribed PKC to the mature (but still inactive) form; phosphorylation is therefore a priming event that enables catalytic activation in response to lipid second messengers. However, many studies have also demonstrated inducible phosphorylation of PKC isoforms at these sites following stimulation, highlighting that our understanding of PKC phosphorylation and its impact on enzymatic function is incomplete. Furthermore, inducible phosphorylation at these sites is often interpreted as catalytic activation."

2. Fig 1B. Not clear what those constructs are; are they mutants? What does 'combine' refer to?
Is the ActX construct active?

- We apologize for the confusion caused in the description of the mutant constructs. The description of these mutants has been changed as follows, and Fig.1A has also been modified so that there is no further confusion. "We created three mutants of PKC β II (ActX1-PKC β II, ActX2-PKC β II, ActX-PKC β II) in which the potential actin binding residues were altered to those of PKC β I. ActX-PKC β II contains all residues altered in ActX1-PKC β II and ActX2-PKC β II."
- We believe ActX is active because it displays some endocytic activity (Fig. 1D), undergoes ubiquitination (Fig. 3C), and PMA-induced phosphorylation at T500 (Fig. 9A).

3. Fig 1C. How many cells counted? Same question for similar results with pictures. Figure 1C and S1C: Additionally, the images are not entirely clear and the LatA condition seems more intense (maybe more gain on the software?). The vehicle +PMA condition morphology looks rather odd compared to the cells presented.

- Five cells were counted both in Fig.1C and Fig.S1C. We noticed that treatment with PMA or LatA, which depolymerizes actin, somewhat causes morphological shape changes. All

the images were captured under the same setting and there has not been any manipulation of the images.

4. Fig 1D and 2D. what is measured? This should be shown on the y axis

→ Thank you for your comments. We changed it to "D₃R Endocytosis (%)"

5. Figure S1D: Does the label KD cells generalize Cav1-KD and CHC-KD? If so there two snips for Cav1 and CHC are different blots and should each have their own appropriate actin control. This validation is important since there is a claim that the endocytosis is Catherin mediated and independent of caveolin.

→ Yes, you are right. Because actin contents were very similar in both KD cells, we have shown just one control. We modified it according to reviewer's comment.

6. Fig 2A. The quantitation does not seem to match the figures. What do the black and grey bars refer to? The quantitation is missing a legend. Cav1-KD +PMA should membrane localize, why does the quantitation show high cytosolic presence?

→ We sincerely appreciate your valuable comments. Data obtained from other cells that were used as controls were mistakenly used for Cav-KD cells. We corrected the graph as the reviewer pointed out.

7. Figure 2B and Others: A minor point for the blots. The actin-GFP + row is not necessary, makes the figures convoluted. Also, not sure why actin is being overexpressed in the cells. Expression of actin is endogenously high, and antibody directed against the protein are specific and tend to give clean blots.

→ Thank you for your comment. Probably you are mentioning Fig.S2 (Fig.S4 in the revised figures). We modified them.

→ As the reviewer pointed out, we tried Co-IP with endogenous actin in the pilot experiments. We could get clean bands with cell lysates, however, for some reasons, only diffuse bands were obtained for Co-IP and we started to transfect GFP-tagged actin.

8. Fig 3A. The labeling of the figure (and many others) is not clear. What is being IP'd? what antibodies are being used for western blots?

→ As described in the ubiquitination assay, the target protein (here PKC β II, FLAG-tagged) was immunoprecipitated and ubiquitin (HA-tagged) was co-immunoprecipitated to detect ubiquitinated PKC β II. We made it clear in the figure legend.

9. Fig. 3 Where (What?) are the higher MW bands in the FLAG-PKC β II? This applies to all panels in this figure.

→ They are ubiquitinated PKC β II. Because PKC β II are polyubiquitinated they display multiple bands in which different numbers of ubiquitins are linked. We made it clear in the figure legend.

10. Fig 3B. the results are not convincing

→ Because PKC β II are polyubiquitinated (Fig. S2), they are displayed as multiple bands on SDS-PAGE. Therefore, ubiquitinated PKC β II cannot help but show diffuse bands rather than one distinct clear band.

11. Fig 3C. results are not convincing.

→ This is the same kind of issue already discussed in #10 above. We conducted three independent experiments and there was statistical significance. In this figure, we added statistical results that we did not add by mistake.

12. Page 10 bottom. Where is the evidence that ubiquitination is needed (allowing) for translocation?

→ This was proven this in a previous publication (Min et al., 2019).

13. Fig 4A. it appears that PDK1 has a modest effect on pPKC T641 (compared with the total pull down).

→ The degree of co-immunoprecipitation was determined by calculating the ratio of co-IP to total IP. We statistically treated based on the measured band densities and there was statistical significance.

14. Figure 5A: Why is there a change in construct used for PKC β II? The previous figures used T500A to assess membrane localization in absence of PDK1 phosphorylation; here PMA treated ubiquitination of PKC β II T500V is assessed instead.

→ It should be T500A. We corrected it both in Fig.4 and Fig.5.

15. Fig 5C. the results are not convincing. What is LMB? What is PCC? Where are results with PMA alone on nuclear localization? What is the Y-Axis and there is no description of this quantitation in the methods or figure captions. Also, it is rather odd that the quantitation of the T500A veh is so high compared to the others and has a large error.

- 'LMB' represents LMB-pretreated cells. 'LMB/PMA' represents LMA pretreatment followed by PMA treatment. LMB is defined in the figure legend.

- Images were processed using Image J software, and colocalization was analyzed based on Pearson's correlation coefficient (PCC, γ value). We made it clear in the figure legend.

- Because PKC β II contents in the vehicle groups are so low, even small dots seemed to affect the basal level.

- In response to the reviewer's comments, we performed subcellular fractionation on these PKC variants (Fig. S3). As shown in these results, all PKC variants except T500A-PKC β II translocated into the nucleus after PMA treatment. These results are consistent with those obtained through confocal studies.

16. Fig 6A. results of the western are not very convincing. Is the 2KR PKC active as a kinase?

→ In my view, co-IP is clearly increased only in the PMA-treated WT-PKC β II group and shows significance when statistical analysis is performed.

- In 2KR-PKC β II, only two lysine residues in the extreme carboxyl tails were mutated to those of PKC β I, which does not undergo PMA-induced ubiquitination (Min et al., 2019). Other than 2KR-PKC β II shows slower translocation to the plasma membrane in response to PMA treatment, 2KR-PKC β II and WT-PKC β II appear to have almost similar properties (Min et al., 2019), suggesting that 2KR-PKC β II is active.

17. Fig 6B. interaction seems somewhat reduced but not absent (conclusion p12: "PKC β II failed to interact, Fig. 6B)

→ We changed it to " The interaction between PKC β II and clathrin was significantly reduced in Mdm2-KD cells compared to Con-KD cells."

18. Fig 6C. the middle images are not clear. The PMA treated conditions may look membrane localized at first glance, but it is hard to tell where the nucleus is and where the cytoplasm is located. This appears as a cell with a high nucleo to cytoplasm ratio, making the sliver of space surrounding the circle be the cytosol and near the plasma membrane.

→ In terms of image quality, I agree with the reviewer's comment. But please consider that it is difficult to catch clear images for endogenous clathrin. In particular, PMA treatment often distorts the shape of cells, making the nucleus look abnormally large.

19. Figure S2A: Lane 4-7 for Actin-GFP look very similar in intensity and do not match the quantitation.

→ We replaced it with a more representative blot (Fig. S4A in the revised manuscript).

20. Figure S2B: Same issue, it is not obvious that the quantitation accurately reflects the actin-GFP co-IP. However, there might be an issue with the image quality that makes it hard to visualize.

→ We replaced it with a more representative blot (Fig. S4B in the revised manuscript).

.

21. Fig 7B. that stats should compare the two PMA treatments for differences.

→ Yes, statistical tests were conducted, but there was no significant difference between the two.

22. Fig 7C. this is more complicated than the text indicates as basal interaction is higher with LMB. For both 7B and C, it may be better to show as a time course similar to A.

→ Reviwer pointed out that the basal interaction was higher in the LMB treatment group in Fig. 7C, but it is not easy to recognize the difference when comparing lanes 2 and 4.

→ The reason we did time-course in Fig.7A is to provide the experimental conditions for Fig.7B and 7C. We fully respect the reviewer's comments, but we believe that Fig.7 conveys enough information as it stands.

23. It would be of high significance to show the phosphorylation of PKC β II on T500 and S641 in panels B and C.

→ It's not clear which figure the reviewer commented on. Phosphorylation of T500 and T641 by PMA treatment is shown in figures such as Fig.8A and 8C.

24. Fig S1C, D: T500V:

→ Thank you for your comments. Probably the reviewer is talking about Fig4. We corrected them.

25. Fig. 9B and C. Does LatA treatment have the same consequence?

→ Yes, essentially the same results were obtained both when cells were treated with LatA and when mutants were used. We decided to use the mutants rather than LatA treatments because LatA makes cell conditions worse.

26. Page 11: phosphatidylserine not phosphoserine

→ Thank you for the comment. We corrected it.

Engel, K., A. Kotlyarov, and M. Gaestel. 1998. Leptomycin B-sensitive nuclear export of MAPKAP kinase 2 is regulated by phosphorylation. *EMBO J.* 17:3363-3371.

Freeley, M., D. Kelleher, and A. Long. 2011. Regulation of Protein Kinase C function by phosphorylation on conserved and non-conserved sites. *Cellular signalling.* 23:753-762.

Lee, S., S. Park, H. Lee, S. Han, J.M. Song, D. Han, and Y.H. Suh. 2019. Nedd4 E3 ligase and beta-arrestins regulate ubiquitination, trafficking, and stability of the mGlu7 receptor. *Elife.* 8.

Min, X., X. Zhang, N. Sun, S. Acharya, and K.M. Kim. 2019. Mdm2-mediated ubiquitination of PKC β all in the nucleus mediates clathrin-mediated endocytic activity. *Biochem Pharmacol.* 170:113675.

- Min, X., X. Zhang, S. Wang, and K.M. Kim. 2023. Activation of PKC β 1 through nuclear trafficking guided by β gamma subunits of trimeric G protein and 14-3-3 ϵ . *Life Sci.* 312:121245.
- Newton, A.C. 2018. Protein kinase C: perfectly balanced. *Crit Rev Biochem Mol Biol.* 53:208-230.
- Shen, T., Y. Liu, Z. Cseresnyes, A. Hawkins, W.R. Randall, and M.F. Schneider. 2006. Activity- and calcineurin-independent nuclear shuttling of NFATc1, but not NFATc3, in adult skeletal muscle fibers. *Mol Biol Cell.* 17:1570-1582.
- Shenoy, S.K., A.S. Modi, A.K. Shukla, K. Xiao, M. Berthouze, S. Ahn, K.D. Wilkinson, W.E. Miller, and R.J. Lefkowitz. 2009. Beta-arrestin-dependent signaling and trafficking of 7-transmembrane receptors is reciprocally regulated by the deubiquitinase USP33 and the E3 ligase Mdm2. *Proc Natl Acad Sci U S A.* 106:6650-6655.
- Tsai, Y.L., Y.C. Mu, and J.L. Manley. 2022. Nuclear RNA transcript levels modulate nucleocytoplasmic distribution of ALS/FTD-associated protein FUS. *Sci Rep.* 12:8180.

January 9, 2023

RE: Life Science Alliance Manuscript #LSA-2022-01748-TR

Prof. Kyeong-Man Kim
Chonnam National University
Pharmacology, College of Pharmacy
77 Yong Bong Ro
Gwangju 500-757
Korea, Republic of (South Korea)

Dear Dr. Kim,

Thank you for submitting your revised manuscript entitled "PKC β II activation requires nuclear trafficking for phosphorylation and Mdm2-mediated ubiquitination". We would be happy to publish your paper in Life Science Alliance pending final revisions necessary to meet our formatting guidelines.

- Please upload all figure files as individual ones, including the supplementary figure files; all figure legends should only appear in the main manuscript file.
- please add a Category for your manuscript in our system
- please add the Twitter handle of your host institute/organization as well as your own or/and one of the authors in our system
- please add your main, supplementary figure, and table legends to the main manuscript text after the references section

FIGURE CHECKS:

- the blot in Figure S3 Nuclear Fraction β -Actin is of poor quality
- please add scale bars to figures 1C, 2A, and S1C, and indicate the sizes of all scale bars in the appropriate figure legends

A. FINAL FILES:

B. MANUSCRIPT ORGANIZATION AND FORMATTING:

Sincerely,

FIGURE CHECKS:

- the blot in Figure S3 Nuclear Fraction β -Actin is of poor quality

→ Since lamin B1, a nuclear marker, and β -actin, a cytoplasmic marker, should be shown under the same experimental conditions, blotting was performed for lamin B1 and β -actin using the remaining sample.

- please add scale bars to figures 1C, 2A, and S1C, and indicate the sizes of all scale bars in the appropriate figure legends

→ We put scale bars in the corresponding figures and also put the sentence "horizontal bars represent 10 μ m" in the corresponding figure legends.

→ We are currently preparing a video clip. We will send it to you as soon as it is completed.

January 11, 2023

RE: Life Science Alliance Manuscript #LSA-2022-01748-TRR

Prof. Kyeong-Man Kim
Chonnam National University
Pharmacology, College of Pharmacy
77 Yong Bong Ro
Gwangju 500-757
Korea, Republic of (South Korea)

Dear Dr. Kim,

Thank you for submitting your Research Article entitled "PKC β II activation requires nuclear trafficking for phosphorylation and Mdm2-mediated ubiquitination". It is a pleasure to let you know that your manuscript is now accepted for publication in Life Science Alliance. Congratulations on this interesting work.

DISTRIBUTION OF MATERIALS:

Again, congratulations on a very nice paper. I hope you found the review process to be constructive and are pleased with how the manuscript was handled editorially. We look forward to future exciting submissions from your lab.

Sincerely,
